



# An organic crystalline state in ageing atmospheric aerosol proxies: spatially resolved structural changes in levitated fatty acid particles

Adam Milsom[1], Adam M. Squires[2], Jacob A. Boswell[2], Nicholas J. Terrill[3], Andrew D. Ward[4] and Christian Pfrang[1,5]

[1] University of Birmingham, School of Geography, Earth and Environmental Sciences, Edgbaston, B15 2TT, Birmingham, UK.
[2] University of Bath, Department of Chemistry, South Building, Soldier Down Ln, Claverton Down, BA2 7AX, Bath, UK.
[3] Diamond Light Source, Diamond House, Harwell Science and Innovation Campus, OX11 0DE, Didcot, UK.
[4] Central Laser Facility, Rutherford Appleton Laboratory, Harwell Campus, OX11 0QX, Didcot, UK.
[5] Department of Meteorology, University of Reading, Whiteknights, Earley Gate, RG6 6BB, Reading, UK.

*Correspondence to*: Christian Pfrang (c.pfrang@bham.ac.uk)

**Abstract.** Organic aerosols are key components of the Earth's atmospheric system. The phase state of organic aerosols is known to be a significant factor in determining aerosol reactivity, water uptake and atmospheric lifetime - with wide implications for cloud formation, climate, air quality and human health. Unsaturated fatty acids contribute to urban cooking emissions and sea spray aerosols. These compounds, exemplified by oleic acid and its sodium salt, are surface active and have been shown to self-assemble into a variety of liquid-crystalline phases upon addition of water. Here we observe a crystalline acid–soap complex in acoustically levitated oleic acid–sodium oleate particles. We developed a synchrotron-based simultaneous Small-Angle & Wide-Angle X-ray Scattering (SAXS/WAXS)/Raman microscopy system to probe physical and chemical changes in the proxy during exposure to humidity and the atmospheric oxidant ozone. We present a spatially resolved structural picture of a levitated particle during humidification, revealing a phase gradient consisting of a disordered liquid crystalline shell and crystalline core. Ozonolysis is significantly slower in the crystalline phase compared with the liquid phase and a significant portion (34 ± 8 %) of unreacted material remains after extensive oxidation. We present experimental evidence of inert surface layer formation during ozonolysis, taking advantage of spatially resolved simultaneous SAXS/WAXS experiments. These observations suggest atmospheric lifetimes of surface-active organic species in aerosols are highly phase dependent, potentially impacting on climate, urban air quality and long-range transport of pollutants such as Polycyclic Aromatic Hydrocarbons (PAHs).



## 1 Introduction

Aerosols are ubiquitous in the atmosphere and are key components of the climate system (Pöschl, 2005; Stevens and Feingold, 2009). They present a large uncertainty when it comes to predicting their effect on the global climate (Boucher et al., 2013). Aerosols can act as pollutants and affect air quality and human health, especially when considering the urban environment (Chan and Yao, 2008; Guarnieri and Balmes, 2014). A large proportion of atmospheric aerosols are organic, (Jimenez et al., 2009) some of which are surface-active (Cheng et al., 2004). Unsaturated fatty acids are a major class of

surface-active organic compounds found in the atmosphere with oleic acid (18-carbon backbone) as a widely-studied example (Gallimore et al., 2017; King et al., 2010; Zahardis and Petrucci, 2007). Sources of atmospheric oleic acid include marine (Fu et al., 2013; Osterroht, 1993) and cooking emissions (Allan et al., 2010; Alves et al., 2020; Ots et al., 2016; Vicente et al., 2018; Zhao et al., 2015). The reaction of organic compounds with the key initiators of atmospheric oxidation: hydroxyl radicals (OH), nitrate radicals ($NO_3$) and ozone ($O_3$) is an important factor in the evolution of these aerosols

(Estillore et al., 2016). Oleic acid, along with the other unsaturated fatty acids, can be oxidised by these species and its reaction with $O_3$ in particular is well-studied and has made it the model system for both theoretical and experimental studies (Gallimore et al., 2017; King et al., 2004, 2009; Last et al., 2009; Milsom et al., 2021; Morris et al., 2002; Pfrang et al., 2017; Schwier et al., 2011; Shiraiwa et al., 2010; Zahardis and Petrucci, 2007).

The phase state of organic aerosols can vary significantly and has been identified as an important factor in

determining atmospheric lifetimes (Shiraiwa et al., 2017; Slade et al., 2019; Virtanen et al., 2010) with particle viscosity being a key property (Reid et al., 2018). Indeed, the chemical lifetime of an organic species in the atmosphere could increase from seconds to days due to temperature and humidity-induced changes in particle viscosity and the diffusion coefficient of molecules through the particle (Shiraiwa et al., 2010, 2011a). Glassy and semi-solid states of organic aerosols have been postulated and studies have shown that a phase transition causes a drastic change in physical properties such as the uptake of

water and reactive gases (Berkemeier et al., 2016; Knopf et al., 2005; Koop et al., 2011; Mikhailov et al., 2009; Zobrist et al., 2011). Diffusion gradients may arise within a viscous organic aerosol particle being exposed to humidity changes and are due to the kinetic limitation of water uptake and loss to and from the viscous particle (Bastelberger et al., 2018; Zobrist et al., 2011). The long equilibration times for these viscous aerosols imply similarly long evolutions in key aerosol properties. Oleic acid, harmful Polycyclic Aromatic Hydrocarbons (PAHs) and phthalates have been identified in marine aerosols that

have been heavily influenced by urban emissions (Kang et al., 2017), suggesting that long-range transport of these molecules does happen. PAH reactivity has been shown to be strongly affected by interactions with particle surfaces (Chu et al., 2010). It has also been indicated that coatings of organic aerosol shield PAHs, increasing their ability to be transported and cause harm and this has been linked to the phase state of the aerosol (Mu et al., 2018; Shrivastava et al., 2017).





As a surface-active molecule oleic acid is able to form, in contact with water, complex self-assembled structures such as organogels (in organic solvents) (Nikiforidis et al., 2015), vesicles (Blöchliger et al., 1998) and even helices (Ishimaru et al., 2005). Mixed with its sodium salt (sodium oleate) and water, oleic acid can also form lyotropic liquid crystalline (LLC) phases (Tiddy, 1980). These phases bring with them a range of different physical properties such as directionally dependent water diffusion, viscosity differences and different optical characteristics. The LLC phase behaviour of oleic acid–sodium oleate has been extensively studied in a biological and cosmetic context and has been shown to have a diverse set of accessible phases ranging from a simple micellar solution through to hexagonal arrays of water channels formed by cylindrical assemblies of the fatty acid (Engblom et al., 1995; Mele et al., 2018; Seddon et al., 1990).

Fatty acids, mixed with their fatty acid soap (salt), can form another set of unique structures called acid–soap complexes (Lynch, 1997). These complexes result from the strong hydrogen bonding between the carboxylate head group of the soap and the carboxylic acid group of the fatty acid and the interactions between the fatty acid alkyl chains. They are stoichiometrically discrete compounds. For the palmitic acid–sodium palmitate acid–soap complex, the sodium ion is shared between adjacent carboxylate anions. Carboxylic acid groups are associated mainly via hydrogen bonding to the carboxylate anions (Lynch et al., 2002). Key properties of acid–soap complexes include: (i) crystallinity: in an atmospheric context this is important, as discussed previously there is a strong link between phase state and the atmospheric properties of an aerosol particle; (ii) unique hydrogen bonding exhibited by their distinct IR spectra (compared to their constituent parts); and (iii) ordered alkyl chain packing deduced spectroscopically and using X-ray techniques such as Small-Angle/Wide-Angle X-ray Scattering (SAXS/WAXS). They are also known to form a range of LLC phases upon addition of water, further demonstrating the versatile nature of the oleic acid–sodium oleate system (Cistola et al., 1986).

Previous work has demonstrated that these LLC phases were present in a levitated unsaturated fatty acid aerosol proxy (Seddon et al., 2016) and that self-assembly drastically reduces the oleic acid ozonolysis reaction rate (Pfrang et al., 2017). The formation of one of these phases (lamellar phase) was found to decrease the ozonolysis reaction rate by *ca.* an order of magnitude (Milsom et al., 2021).

In this work, the importance of the oleic acid–sodium oleate acid–soap complex in atmospheric conditions is investigated. This complex has previously been studied in a biological context (Ananthapadmanabhan and Somasundaran, 1988; Tandon et al., 2001). The hydrocarbon chain and head group of the fatty acid can be characterised by complementary Raman and IR spectroscopy: the acid–soap complex has characteristic peaks both in the IR and Raman spectra, allowing confirmation of the structure of the complex (Lynch et al., 1996; Tandon et al., 2001). SAXS and WAXS have also been used to confirm the lamellar packing of the acid–soap complex and to reveal the sub-cell packing arrangement of the alkyl chains unique to the acid–soap complex (Tandon et al., 2001). We also employed Polarising Optical Microscopy (POM) in order to visualise structural changes with temperature and humidity.

The oleic acid–sodium oleate acid–soap complex is studied in acoustically levitated droplets and analysed by simultaneous SAXS/WAXS and Raman microscopy. Ozonolysis is followed by Raman, while the effect of oxidative ageing on self-assembly is investigated using SAXS/WAXS. We first carried out a detailed structural characterisation of the acid–





soap complex to confirm its presence in the levitated particles. We then probed the effects of exposure to humidity and ozone. We employed a micron-sized X-ray beam (16 µm x 12 µm) available on the I22 beamline at the Diamond Light

Source (UK) to follow structural changes throughout the particle during controlled humidity changes and atmospheric ageing. This enabled us to build a spatially resolved SAXS/WAXS picture of the particle as a function of time exposed to humidity and ozone. Using this technique, we observed the emergence of a diffusion gradient across the humidifying/dehumidifying acid–soap particle and described this effect. We also investigated if exposure to ozone destroys the acid–soap structure and if the crystalline structure affects the reaction kinetics, drawing atmospheric implications.

**2 Methodology**

A 1:1 wt ratio of oleic acid and sodium oleate was chosen to afford a system with a molar excess (7.8 %) of oleic acid, simulating the acidic nature of urban (Zhang et al., 2007) and marine (Keene et al., 2004) aerosols.

**2.1 Sample preparation**

Oleic acid ((Z)-octadec-9-enoic acid, 90 %) and sodium oleate (sodium (Z)-octadec-9-enoate, 99 %) were purchased from

Sigma-Aldrich (UK) and used as received. Oleic acid and sodium oleate were weighed in a 1:1 wt ratio and dissolved as an ethanolic solution in a minimum of hot ethanol. This solution was allowed to cool to room temperature (~ 22 ℃). The ethanol in the sample was evaporated before being placed into an acoustic node for the study of levitated particles. Samples were alternatively deposited on glass microscope slides for offline Raman Microscopy and POM.

**2.2 Preparation of the bulk oleic acid/sodium oleate/water mixture**

Oleic acid and sodium oleate were weighed and mixed in a 1:1 wt ratio with addition of water in order to afford a final mixture which is 70 % aqueous phase. The mixture was homogenised first by sonicating in a heated ultrasonicator (~ 40–50 ℃) for 30 min, followed by vortexing for 1 min. The sample was then placed in a freezer for storage and further homogenisation. Upon defrosting, the sample was observed to be homogeneous, as subsequently confirmed by the SAXS pattern.

The sample was then placed inside a poly-imide tube. The tube was sealed at both ends with heat-shrink tubing. The sample was then placed in the beam path of an *Anton Paar SAXSpoint 2.0* instrument at the University of Bath. The sample was irradiated by X-rays with a 1.54 Å (Cu source) wavelength for 3 min at a 360 mm sample-detector distance.

**2.3 Simultaneous Raman microscopy and small-angle/wide-angle X-ray scattering (SAXS/WAXS) of levitated particles during exposure to humidity and ozone**

A modified commercial levitator (tec5, Oberursel, Germany) with a fixed transducer frequency (100 kHz) and variable HF power (0.65–5 W) was used to levitate the atmospheric aerosol proxies. A concave reflector was positioned above the





transducer and was fitted with a micrometre screw to adjust the reflector–transducer distance. The reflector–transducer distance was generally in the range of 20–30 mm. The levitator was enclosed in a 3–D-printed chamber equipped with X-ray transparent Mica windows and access ports for injection, Raman probe and gas in- and outlets.

The levitated particles were analysed on the I22 beamline at the Diamond Light Source (UK). Solid samples, crystallised from ethanolic solutions, were placed into a node of the acoustic levitator. The particles had vertical radii of ~ 90–150 µm and horizontal radii of ~ 500 µm (determined using the attenuation of the X-ray beam). Once the particle stabilised, a 532-nm Raman laser probe with a 12 mm focal length and ~ 4 µm focal point spot diameter (minimum) was focussed onto the particle. The laser power delivered to the particle was determined to be ~ 20 mW (source power up to 450

mW).

A dry flow of oxygen was passed through a commercial pen-ray ozoniser (Ultraviolet Products Ltd, Cambridge, UK). The ozone concentration was kept constant at 51.9 ± 0.5 ppm and was calibrated by UV/Vis spectroscopy using the ozone absorption band at 254 nm and the absorption cross-section for ozone at this wavelength ($1.137 \pm 0.070 \times 10^{-17}$ cm$^2$) (Mauersberger et al., 1986). The high ozone concentration was chosen to observe an appreciable decay within the limited

timescale of a synchrotron experiment.

SAXS/WAXS patterns were collected as a series of 1 s frame vertical scans across the particle. There was an intentional delay of 15 s between each set of scans to avoid any potential X-ray beam damage. A micro-focus X-ray beam was used to enable sufficient spatial resolution of the SAXS patterns within the particle. The size of the micro-focus beam was approximately 16 µm x 12 µm (FWHM). Scattering patterns up to $q = 0.58$ Å$^{-1}$ were recorded by the SAXS detector (*Pilatus*

*P3-2M*) and from 0.50–4.45 Å$^{-1}$ by the WAXS detector (*Pilatus P3-2M-L*). The scattering intensity is related to the momentum transfer ($q$), which is a function of scattering angle and is related to the spacing ($d$) between scattering planes via Eq. (1) (Putnam et al., 2007).

$$q = \frac{2\pi}{d} \tag{1}$$

A bespoke relative humidity (RH) control system, using a Raspberry Pi with air pumps and RH sensor, was used to monitor

and control RH in real-time. The line of humidified air was passed into the levitation chamber. An RH/temperature sensor was placed inside the chamber for real-time monitoring of the chamber RH and temperature.

### 2.4 Offline Raman microscopy

A *Renishaw InVia* Raman microscope was used to analyse samples deposited onto microscope slides. This was achieved by placing a drop of ethanolic solution on the slide and allowing it to evaporate in air. Cool air was passed over the sample to

aid evaporation. A film was left deposited on the microscope slide. A 532-nm laser was focussed onto the sample using a 20x objective lens and Raman spectra were acquired in the range 100–3500 cm$^{-1}$ with two acquisitions.





## 2.5 Polarising optical microscopy (POM)

All POM was performed using a *Carl Zeiss Axioskop 40* fitted with removable polarising filters. Samples were prepared as for the Raman microscope. Visualisation was accomplished using either 5x or 10x objective lenses. A Peltier heating stage
was used to control the temperature of samples under the microscope.

Samples deposited on microscope slides were humidified by suspending the slides above distilled water inside a small, sealed, container. This provided a saturated environment for the samples to equilibrate with.

## 2.6 Infrared spectroscopy (IR)

Infrared spectroscopy was carried out on a *PerkinElmer Spectrum 100* FTIR spectrometer with an ATR attachment.
Measurements comprised of 32 scans at a resolution of 4 cm$^{-1}$. This technique required more material to analyse, therefore a small amount of ethanolic sample solution was left to evaporate to leave behind crystals. The quantitative evaporation of ethanol in the samples was confirmed by IR spectroscopy. All samples were analysed at room RH of ~ 50 %.

## 3 Results and discussion

### 3.1 Characterisation of the acid–soap complex by SAXS/WAXS

The dry acid–soap complex was probed by simultaneous SAXS/WAXS and Raman in acoustically levitated droplets. SAXS allows investigation of the long-scale order (repeated structures). The regular interval between each SAXS peak in Fig. 1(a) is characteristic of a lamellar system with a *d*-spacing of $4.5773 \pm 0.0001$ nm between repeat structures (see Table S1).

The WAXS data (Fig. 1(b)) reveal information about the packing of the alkyl chains in particular and are consistent with the literature data at a lower temperature (Tandon et al., 2001). A table of WAXS data and a comparison with the
literature is presented in the Supplement (Sect. S1). They also confirm the crystallinity of the sample. Note the broad peak underneath the sharp WAXS peaks in Fig. 1(b) is due to the 7.8 % molar excess of oleic acid (with its own WAXS peak – see Sect. 3.3.2) in the sample. A fuller characterisation involving Raman microscopy, IR spectroscopy and polarising optical microscopy (POM) is presented in the Supplement (Sect. S1).





**Figure 1.** ((a) and (b)) 1-D SAXS and WAXS patterns obtained from a dry levitated particle of the acid–soap complex – 1st, 2nd and 3rd lamellar peaks are labelled and a cartoon of the lamellar phase is presented (a). ((c) and (d)) experimental fraction of maximum water content as a function of distance from particle centre and time humidifying/dehumidifying. ((e) and (f)) modelled fraction of maximum water content – best fits to experimental data for humidification and dehumidification. 3–D surface plots of 1–D SAXS patterns plotted against distance from the particle centre for the same particle humidifying ((g) – (i)) and dehumidifying ((j) – (l)) with time humidifying/dehumidifying presented at the top right of each plot (particle size: ~ 150 μm (vertical radius) x 500 μm (horizontal radius); humidification experiment: ~ 38 % to 90 % RH, dehumidification experiment: 90 % to ~ 38 % RH).





### 3.2 Atmospheric ageing: (i) exposure to humidity changes

Field measurements have shown atmospheric aerosols to be often in a highly viscous state (Virtanen et al., 2010). Aerosol particle viscosity is dependent on water content, which in turn is controlled by its chemical nature and the variable surrounding relative humidity (Fitzgerald et al., 2016; Hosny et al., 2016; Renbaum-Wolff et al., 2013; Shiraiwa et al., 2011a). Changing the humidity of the sample environment effectively controls how much water is taken up by the particle. Therefore it is necessary to build an understanding of this system's behaviour when subjected to atmospherically relevant humidity changes.

The acid–soap complex studied here is crystalline and as such, may have different hygroscopic properties compared to liquid aerosols. The literature presents a conceptual framework for the interaction of amorphous and crystalline aerosol particles with water (Koop et al., 2011; Mikhailov et al., 2009). It suggests that crystalline particles deliquesce promptly whereas amorphous particles gradually absorb water and deliquesce at a slower rate due to limited water diffusion through the amorphous phase.

### 3.2.1 Structural changes during humidification and dehumidification

Our humidity–SAXS/WAXS experiments on levitated acid–soap complex particles presented here suggest that these particles do take up water at high humidity. The levitated particle exhibited reversible structural changes as a result of one humidification–dehumidification cycle, illustrated in Fig. 1(g)-(l) by the disappearance and reappearance of the acid–soap complex SAXS peaks during humidification and dehumidification. Simultaneous SAXS/WAXS data from the particle centre are presented in the Supplement (Fig. S4). The horizontal position was changed by 15 µm between the humidification and dehumidification runs in order to check for beam damage, of which there was no evidence (see Fig. 1(g)-(l) and Fig. S4 – SAXS patterns are identical between the last humidification and first dehumidification runs).

The particles were exposed to an RH of 90 % and SAXS/WAXS patterns were collected. Water uptake changes the physical characteristics of the droplet and therefore small size and shape changes would occur (Mikhailov et al., 2009), resulting in some destabilisation of the levitated particle. Such physical changes were observed visually for a particle of sodium oleate offline (Fig. S7).

The acid–soap complex breaks down when exposed to a high-humidity environment. The dry acid–soap sample has a lamellar SAXS pattern with characteristic peaks in the WAXS (Fig. 1(b)). Upon adding humid air, a small broad peak at ~ 0.2 Å$^{-1}$ appeared almost immediately (Fig. 1(g) and Fig. S4). This broad peak is caused by the formation of a disordered inverse micellar phase. This phase becomes more prominent during continuous exposure to 90 % RH. Eventually, there is a decrease and then disappearance of the acid–soap lamellar phase signal in the SAXS data. This is also true for the WAXS peaks (Fig. S4).

Dehumidification mostly reversed this trend, though a broad inverse micellar peak still remained present by the end of the experiment – evident in the SAXS pattern (see Fig. 1(l) and succeeding discussion).





This experiment demonstrates that the acid–soap complex can form from an inverse micellar phase of this particular

composition in water. As it is highly unlikely for there to be enough ethanol in the atmosphere to form this complex by evaporation from an ethanolic solution, the most likely atmospheric formation pathway is from an aqueous inverse micellar phase of this composition, as presented here.

Spatially resolved 1–D SAXS patterns were acquired whilst scanning through the particle during exposure of the acid–soap complex to high RH (Fig. 1(g)-(l)). At the beginning of the humidification experiment, sharp, evenly spaced peaks

were visible in the SAXS (Fig. 1(g)). This is consistent with the lamellar packing of the acid–soap complex. Moving through the particle, the broad inverse micellar peak is approximately the same intensity relative to the most intense first order lamellar peak. The increase in overall intensity approaching the centre of the particle is expected as the X-ray beam travels through more material and therefore more scattering occurs. In general, the particle shows a consistent composition throughout. The characteristic acid–soap complex WAXS peaks are also observed and confirm the crystalline nature of the

particle at the beginning of the experiment (Fig. S4).

During humidification the broad inverse micellar peak present at the beginning of the experiment becomes more intense and the sharp lamellar signal starts to disappear, starting from the edges of the particle (Fig. 1(h)). For example after 226 min, there is an apparent difference in SAXS patterns between the edge and centre of the particle. The edge region exhibits only the broad disordered inverse micellar peak, suggesting that the uptake of water breaks the acid–soap complex

down into this disordered phase. By the end of the hydration experiment the SAXS pattern throughout the particle consisted of one broad inverse micellar peak (Fig. 1(i)). This shows that the particle had taken up water throughout and that the acid–soap complex was no longer present. A similar change was seen in the WAXS pattern (Fig. S4).

The reverse trend is observed during dehumidification, though the phase change from inverse micellar to lamellar particle happens markedly faster (Fig. 1(j)-(l)) and Fig. S4). It is clear that the crystalline lamellar phase signal is most

intense in the centre of the particle during dehumidification. There is evidence of the lamellar phase forming on the outside of the particle, suggesting that the phase change is spatially more uniform than during humidification (Fig. 1(k)).

Figure 1*H* illustrates the presence of a viscous (crystalline lamellar) core with a less viscous (inverse micellar) shell, inferring that a diffusion gradient is established during particle humidification (see Sect. 3.2.2). Diffusion gradients have been theorised in the past when humidifying atmospherically relevant viscous systems (Bastelberger et al., 2018; Zobrist et

al., 2011). This kind of core-shell morphology is plausible for particles of this size (Veghte et al., 2013).

The inverse micellar phase and acid–soap complex will have different viscosities and as such, reactive gas and water uptake may be significantly affected. The literature suggests that at high RH, moisture-induced phase transitions could increase reactive gas uptake by reducing the viscosity of the aerosol particle (Shiraiwa et al., 2011). Hosny *et al.* visualised this viscosity gradient using a fluorescence-based technique to probe the viscosity response of oxidised organic aerosols to a

step change in RH (Hosny et al., 2016). See Sect. 4 for further discussion.

The acid–soap complex scattering pattern was observed after dehumidification (Fig. 1(j)–(l)). There was, however, a broad inverse micellar peak present at the end of dehumidification. The suggestion is that a phase separation took place





within the particle – the two phases being the crystalline acid–soap complex and an inverse micellar phase. Previous work has focussed on liquid–liquid phase separations at high RH in organic aerosols (Freedman, 2017, 2020; Liu et al., 2018). In this study, a water-containing inverse micellar phase forms initially on the outside, creating a clear phase gradient during humidification. Upon dehumidification of the levitated particle from 90 % RH to ~ 38 % RH, some of the inverse micellar phase remains in the particle. The SAXS pattern is consistent throughout the particle which suggests that this phase separation is not due to some inverse micellar phase being trapped within the particle as a result of the rapid dehumidification. Rather, the dehumidified particle may have the two phases evenly distributed throughout.

Long-term humidity exposure experiments were performed on acid–soap complex samples deposited on microscope slides. POM of these samples revealed an eventual transition to the inverse hexagonal phase after a week of humidification followed by phase separation to an acid–soap complex and a non-birefringent (not lamellar or inverse hexagonal) phase after removal from the humid environment (Fig. S9).

### 3.2.2 The water diffusion gradient during humidity change

In order to estimate the water diffusion gradient between the inverse micellar and crystalline lamellar phase during humidification and dehumidification, a suitable parameter corresponding to water content was required. The position (in $q$) of the micellar peak centre in the SAXS pattern was chosen for the dehumidification experiment as this is inversely proportional to the micellar $d$-spacing $i.e.$ the average distance between inverse micelles (Eq. 1), which is determined by the water content of the inverse micellar phase. For humidification, the micellar-lamellar peak area ratio was chosen as a suitable measure of water content. In this case $q$ is not a good measure due to the peak position stabilising after ~ 50 min (Fig. S4), implying that water uptake finishes at that time when it is clear from the micellar-lamellar peak area ratio data that water uptake had not finished (Fig. S10). This is due to the inverse micellar phase reaching an equilibrium $d$-spacing, from that point onward the peak area increases while the peak position stays the same.

Water concentration was not quantified as this would have required determining the dependence of $d$-spacing on water content, which was not practicable during a beamtime experiment and would require water content to be measured over a very small range – increasing the amount of water can also change the self-assembled phase (Engblom et al., 1995; Mele et al., 2018). However, there is confidence that the water content is ~ 5 wt % at high RH as previous literature (in dilute salt solution) suggests the inverse micellar phase forms at this water content before becoming an ordered inverse micellar phase at 10 wt % (Mele et al., 2018). Water content was measured as a fraction of the maximum micellar-lamellar/$d$-spacing peak area ratio observed in the particle during the entire humidification/dehumidification experiment.

A simple model of water uptake and loss was created to account for experimental observations. The particle was split into layers, equivalent to the number of positions probed across the particle. Water uptake into/out of the particle and water diffusion between model layers were described by two parameters: rate in/out of the particle ($k_{in/out}$) and rate of internal diffusion ($k_{internal}$), which was split into the rate of water diffusion in the inverse micellar phase ($k_{micellar}$) and crystalline lamellar phase ($k_{lamellar}$). Splitting $k_{internal}$ in this way, and assuming the direct proportionality of $k_{micellar}$ and $k_{lamellar}$ with their





respective diffusion coefficients, allowed parameters for internal diffusion to be evolved as a function of layer composition using a Vignes-type equation (see Sect. 8 of the Supplement) (Davies and Wilson, 2016; Price et al., 2015). See the Supplement for a schematic representation of the model (Fig. S11).

As actual water content data were not available for these experiments, water content as a fraction of maximum water content was used in the model in order to fit with the experimental data. Note that water content data derived from micellar–lamellar peak area ratios is noisier than those derived from $d$-spacing measurements.

During humidification the model first overpredicts then underpredicts the amount of water in the particle (Fig. 1(c) and (e)). Clearly there is very little water observed experimentally in the centre of the particle at ~ 200 min (~ 1 % maximum water content), whereas the model returns ~ 16 % maximum water content. After ~ 230 min humidification the water content

of the whole particle increases sharply, in line with the crystalline model for water uptake (Mikhailov et al., 2009). The model now underpredicts the experiment, peaking at ~ 40 % maximum water content whereas the experimental data tends to 100 %. The uptake model does not capture this sudden jump in water content. We suggest that this is due to the restructuring and prompt deliquescence of our crystalline particle, not accounted for in the model and described by Mikhailov *et al* (Mikhailov et al., 2009). This finding supports the crystalline lamellar core–inverse micellar shell observed in the

corresponding spatially resolved SAXS patterns (Fig. 1(h)).

During dehumidification the model agrees well with the experiment for the first ~ 65 min (Fig. 6(d) and (f)). After ~ 65 min the acid–soap complex signal starts to dominate the SAXS pattern (Fig. 1(k) and Fig. S4). The more viscous acid–soap complex phase slows down water diffusion. The final experimental and model water contents are ~ 17 % and ~ 5 % maximum water content, respectively. There is still an appreciable difference between model and experiment, likely

reflecting the irreducible noise in the experimental data.

The model was parameterised in order to estimate the relative water diffusivity in the inverse micellar phase compared with the crystalline lamellar phase. We used the dehumidification experiment and model fit to estimate these values as this experiment returned the least noisy data and the lowest fitting error (see Sect. 8 of the Supplement). Water diffusivity was found to be ~ 33-fold greater in the inverse micellar phase. We stress that this is an estimation based on our

simplified model of water uptake and diffusion with limited knowledge of the actual water content. Increasing the complexity of the model would introduce too many unknowns that cannot be constrained with currently existing experimental data. This does however open the door to future, more explicit, descriptions of water diffusivity changes in self-assembled systems, analogous to studies of ultraviscous aerosols (Davies and Wilson, 2016; Price et al., 2015; Zobrist et al., 2011). See Sect. 4 for a discussion of the significance of these findings.

**3.3 Atmospheric ageing: (ii) exposure to ozone**

Levitated particles of the acid–soap complex were exposed to ozone in order to simulate chemical ageing of the particle in the atmosphere. Ozone attacks the carbon–carbon double bond found half way along the alkyl chain. A complex reaction mechanism ensues involving Criegee intermediates. Major products include nonanal, nonanoic acid, 9-oxo-nonanoic acid





and azaleic acid (Gallimore et al., 2017; Hung et al., 2005; Zahardis and Petrucci, 2007). Oligomeric products have also been

observed and result from the reaction of Criegee intermediates with the reaction products mentioned above (Reynolds et al., 2006; Zahardis et al., 2006).

Water is known to affect the reaction mechanism of ozonolysis and affect the product distribution in oxidised oleic acid particles (Al-Kindi et al., 2016; Vesna et al., 2009). In our experiments, ozonolysis was carried out under dry conditions (< 5 % RH) in order to negate any effect the presence of water might have on the ozone uptake and reaction rate (Berkemeier

et al., 2016; He et al., 2017; Nájera et al., 2015).

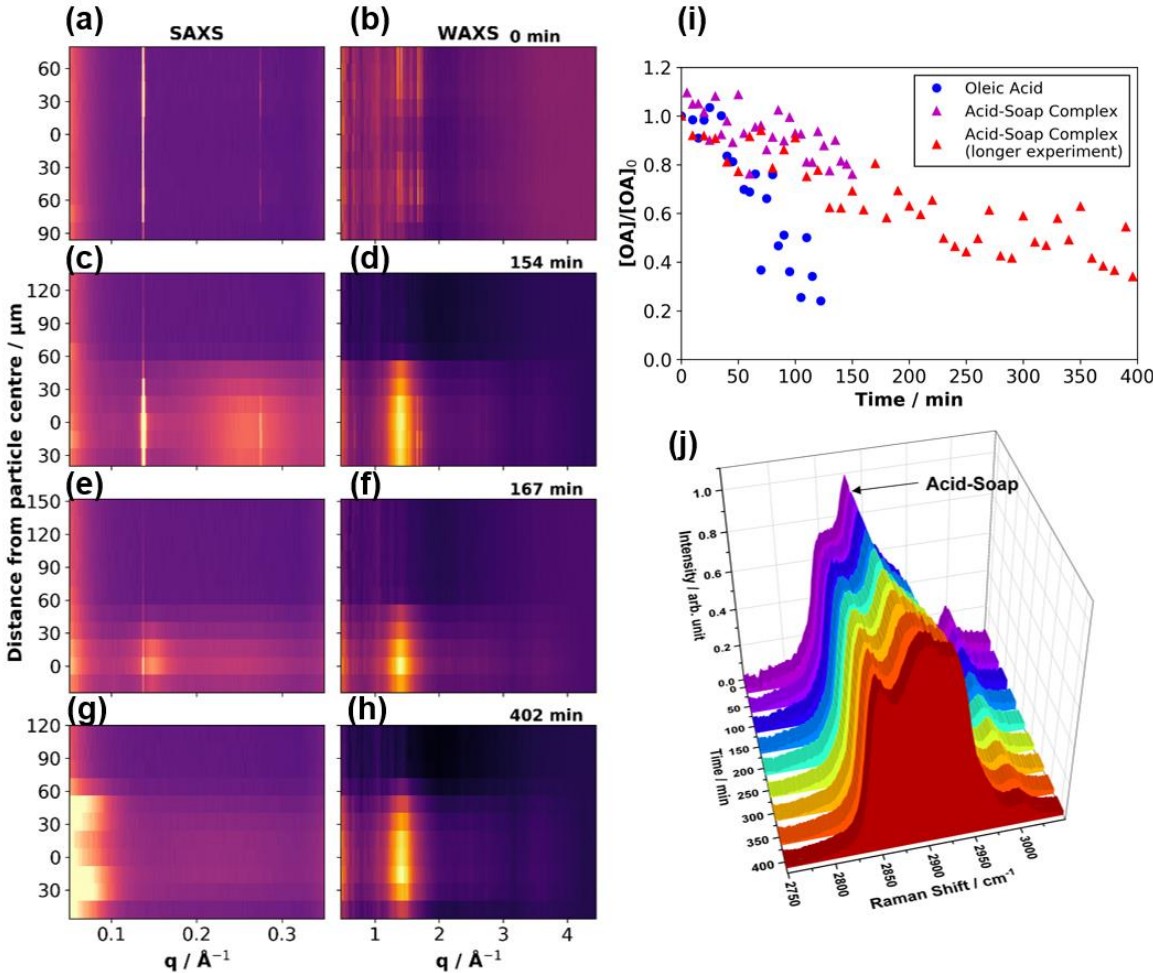





**Figure 2. Vertical scans through the particle showing the effect of ozonolysis on self-assembly. Each row of plots ((a) and (b), (c) and (d), (e) and (f), (g) and (h)) shows simultaneous 1–D SAXS and WAXS scattering patterns *vs.* distance from the particle centre (measured in µm from what was deemed the particle centre from attenuation data) at increasing time exposed to ozone (labelled at the top-right of every WAXS plot). The particle moved and possibly changed shape during the experiment, vertical movement is apparent from the SAXS and WAXS patterns. (i) Comparison of a levitated pure oleic acid droplet *vs.* a levitated acid–soap complex particle undergoing ozonolysis, measured by Raman microscopy - A longer ozonolysis experiment on a different levitated acid–soap complex particle is also presented. (j) Evolution of the Raman spectra between 2750 and 3050 cm$^{-1}$ of a levitated acid–soap complex during ozonolysis. (Particle size: ~ 85 µm (vertical radius) x ~ 500 µm (horizontal radius); [O$_3$] = 51.9 ± 0.5 ppm).**

### 3.3.1 The effect of the crystalline phase on reactivity

After considering the effect of humidity, we now explore specifically the effect of phase on chemical kinetics in dry conditions with a focus on spatial resolution of the phase evolution across individual droplets, which is the key strength of our experimental approach. The impact of humidity on chemical ageing will be discussed in the *Atmospheric Implications* and explored by follow-on modelling work since it cannot be de-convoluted by experimental work alone given the complex interplay of humidity, phase and chemistry.

Kinetics are followed by Raman microscopy. The area of the C=C peak (~ 1650 cm$^{-1}$) is integrated and normalised against the –CH$_2$ deformation band (~ 1442 cm$^{-1}$). A decay plot (Fig. 2(i)) is then created normalising to the starting C=C/–CH$_2$ ratio. The Raman spectra from these beamline experiments had a high and varying background and as such the signal-to-noise ratio was poorer compared to experiments carried out offline (Fig. S6).

The two particles used for ozonolysis experiments were non-spherical with vertical and horizontal diameters of ~ 170 x 1000 µm and ~ 225 x 1000 µm as determined from our SAXS data. An optical picture (taken offline) of a levitated particle of sodium oleate is presented in the supplement as an illustration of the particle shape (Fig. S7).

Figure 7 demonstrates that the levitated particles of the acid–soap complex are much less reactive than droplets of oleic acid. The ratio of reactivity between oleic acid vs the acid–soap complex is 4.95 ± 0.40, suggesting that oleic acid in the crystalline acid–soap complex form reacts significantly slower than in the liquid form. This is consistent with observations previously made on levitated complex 3–D self-assembled aerosol proxies.(Pfrang et al., 2017) We are now able to quantify impact of phase on the reactivity of oleic acid.

The C=C peak at ~ 1650 cm$^{-1}$ does not disappear entirely by the end of the reaction and 34 ± 8 % of oleic acid remains in the particle, suggesting there is still unreacted oleic acid at the end of the experiment despite being exposed to a high [O$_3$] (51.9 ± 0.5 ppm) for more than 6 hours (Fig. 2(i)).

### 3.3.2 Evolution of the SAXS pattern during ozonolysis

The SAXS pattern of the levitated acid–soap complex during ozonolysis evolved slowly thoroughout the particle. Initially, the particle is lamellar throughout and the characteristic acid–soap complex WAXS peaks are present (Fig. 2(a) and (b)). As the reaction progresses, broad features start to appear in the SAXS pattern at ~ 0.15 Å$^{-1}$ and 0.27 Å$^{-1}$ close to the original lamellar peaks and the original WAXS signals start to fade, with a new broad peak starting to appear at ~ 1.4 Å$^{-1}$ (Fig. 2(c) and (d), (e) and (f)). These features are due to the gradual disordering of the crystalline lamellar structure. The broad WAXS





peak corresponds to an average spacing between alkyl chains of ~ 4.52 Å which is similar to the value we obtained for pure

oleic acid at ~ 4.57 Å (Fig. S13). There is also a similarity to the value measured by Iwahashi *et al.* (4.58 Å) (Iwahashi et al.,

1991) which is associated with dimer formation in "free" oleic acid. By the end of the reaction the lamellar phase signal in

the SAXS pattern has disappeared and the broad WAXS peak associated with oleic acid remains (Fig. 2(g) and (h)).

An increasing amount of low-$q$ SAXS scattering is observed by the end of the reaction (Fig. 2(g)). $q$ is inversely

proportional to the distance between equivalent scattering locales. Low-$q$ scattering therefore implies that molecules which

exhibit some order with relatively large repeat distances  have been formed (note there is always some background scattering

at low-$q$ close to the X-ray beamstop). This low-$q$ scattering may come from material such as an oligomeric/high-molecular-

weight product, which has been observed as a result of oleic acid ozonolysis (Lee et al., 2012; Reynolds et al., 2006; Wang

et al., 2016; Zahardis et al., 2005, 2006). Water reacts with Criegee intermediates (Vesna et al., 2009), which are involved in

the formation of these high molecular weight oligomers. These ozonolysis experiments were carried out under dry (< 5 %

RH) conditions, raising the probablility of oligomer formation. Further discussion of the evidence for oligomer formation is

presented in Sect. 6 of the Supplement. It is evident from the vertical SAXS/WAXS profile (Fig. 2(g) and (h)) that the oleic

acid WAXS signal was most intense at the centre of the particle and least intense at the edges. Significantly, low-$q$ SAXS

signals are more intense towards the edges of the particle, suggesting that there is a shell of oligomeric/high-molecular-

weight material encompassing a free oleic acid core – though weaker WAXS signals are still observable at the edges, which

suggests that free oleic acid is also present there to some degree - diffusion of which would be impeded by the viscous layer.

Raman spectra suggest that a significant amount of oleic acid does indeed remain after oxidation, supporting the structural

findings presented here (see Sect. 3.3.3). The significance of this core–shell finding is discussed in Sect. 4.

These experiments provided ~ 16 x 12 µm resolution SAXS images across the particle. This is the first time that

spatially resolved structural changes have been measured in acoustically levitated particles during ozonolysis. This

resolution gives a structural insight into the evolution of a particle during ozonolysis, allowing for the first time to draw a

time-resolved self-assembled phase picture across a particle.

### 3.3.3 Evolution of the Raman spectrum during ozonolysis

Three key changes in the Raman spectrum are observed during ozonolysis. First, there is a clear shift of the strong acid–soap

peak from ~ 2887 cm$^{-1}$ to ~ 2897 cm$^{-1}$ accompanied by some broadening. This is indicative of the loss of alkyl chain order

upon degradation of the acid–soap complex (Tandon et al., 2001). Secondly, the weak shoulder at ~ 2854 cm$^{-1}$ becomes

more defined during oxidation. This is further evidence, in combination with SAXS observations (Fig. 2(j)), that the oleic

acid left in the system is not involved in an acid–soap structure after ozonolysis. This also stresses the importance of the

simultaneous SAXS/Raman technique for time-resolved structural and chemical analysis. Similar changes in the Raman

spectra were observed during week-long high humidity experiments, in which the acid–soap complex also breaks down due

to the formation of liquid crystalline phases in which the hydrophobic tails are not well-packed (Fig. S9). Finally, the C=C-H

peak at ~ 3000 cm$^{-1}$ decreases in intensity due to the loss of unsaturation and removal of oleic acid from the system. It is



important to note that this peak, along with the C=C peak at ~ 1650 cm$^{-1}$ does not disappear entirely by the end of the reaction.

The observed evolution of the SAXS/WAXS and Raman data during ozonolysis coincide with each other and are complementary to one another since they follow structural and chemical changes simultaneously, demonstrating the power of the SAXS/WAXS/Raman technique for investigation of levitated particles. Note that the spatial scale is different between the techniques: micro-focus SAXS/WAXS experiments used a beam width and height of ~ 16 x 12 µm whereas the Raman laser spot diameter was ~ 4 µm focussed on the bulk of the particle. Both techniques concurrently confirm that the acid–soap

complex breaks down as a result of simulated atmospheric ageing by ozone, however the Raman spectrum clearly demonstrates that 34 ± 8 % of oleic acid -generally assumed to be broken down efficiently by ozone- remains after the oxidation process. The significance of this is discussed in the following section.



## 4 Atmospheric implications

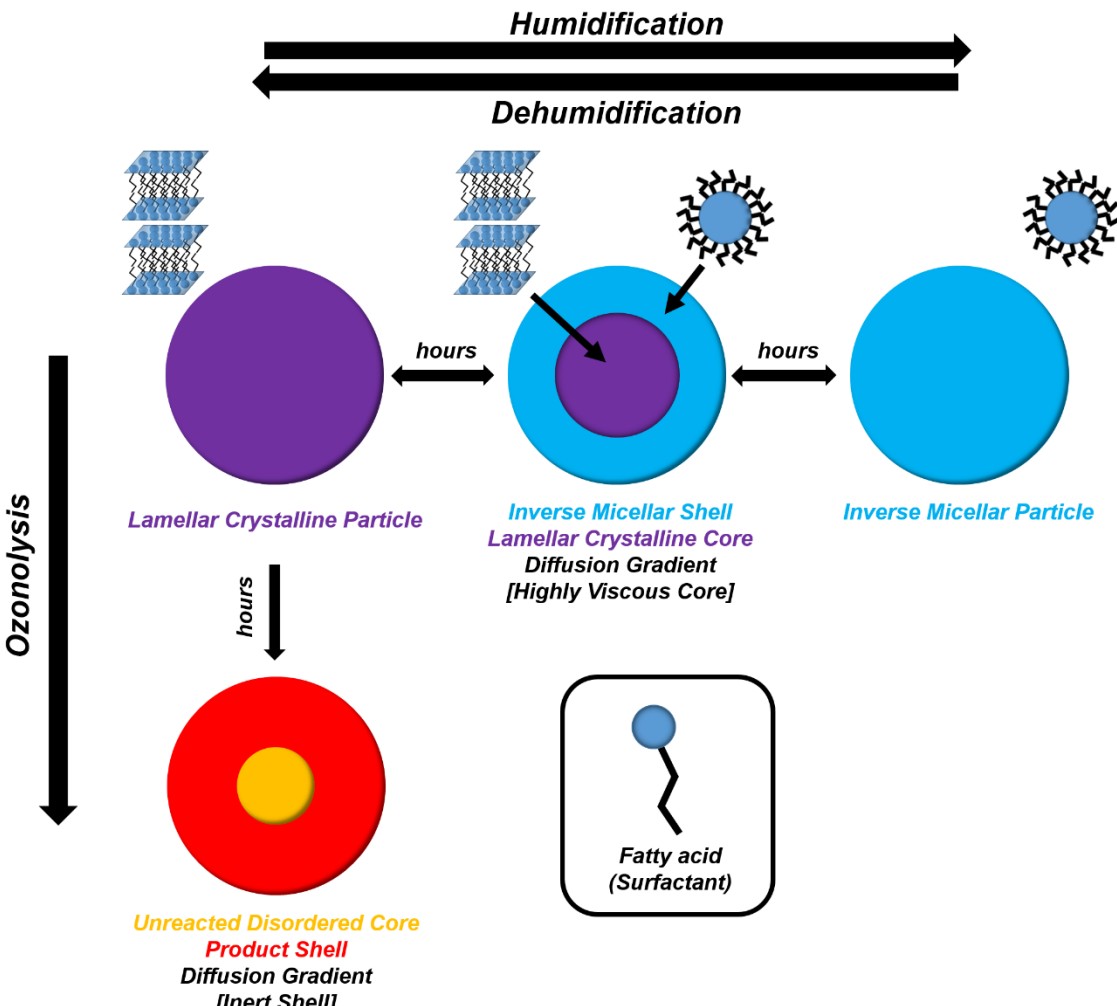

**Figure 3. A schematic summarising the findings of this study. Particle humidification-dehumidification and ozonolysis are represented showing core–shell behaviour. Cartoons of SAXS-observed phases are labelled for clarity.**

The acid–soap complex reported here is a crystalline organic material that is made up of surface-active molecules. The observations in this study suggest that water uptake is in line with the crystalline water uptake model described in the literature (Koop et al., 2011; Mikhailov et al., 2009). The addition of water gradually broke down the acid–soap complex and created a majority inverse micellar phase initially on the outside of the particle, then throughout the particle by the end of the experiment. This inverse micellar phase will have a different bulk diffusion coefficient compared to the crystalline solid (Shiraiwa et al., 2011a). Our results suggest that water uptake into these separate crystalline and liquid crystalline phases is markedly different due to viscosity-related differences in diffusivity. Equilibration times are longer than what would be



expected for a well-mixed liquid particle, explaining the observation of a water diffusion and particle phase gradient throughout the humidifying/dehumidifying particle (Fig. 1(c)-(f) and (g)-(l)). A similar trend is expected for other

atmospheric trace gases such as ozone, implying the reactive lifetime of this aerosol would be affected by water uptake. These observations were backed up by the use of a simple model of water uptake and loss, which revealed that water diffusivity is reduced by ~ 33-fold in the crystalline lamellar phase compared with the inverse micellar phase.

Previous work has not considered the formation of inverse micelles, though there has been a theoretical study on aggregate formation in organic aerosols where the formation of micelles was postulated (Tabazadeh, 2005). However, recent

atmospheric (Pfrang et al., 2017) and biological (Mele et al., 2018) studies into the oleic acid/sodium oleate/water self-assembled system have shown that the *inverse* micellar phase can form for this system due to a significant non-polar "oily" fraction in the mixture (*i.e.* oleic acid). This is because the majority of head groups in the mixture are protonated (*i.e.* uncharged). It is therefore likely that the micellar phase observed in this system has an inverse rather than the normal topology suggested in the preceding atmospheric literature (see Fig. 3 for a cartoon representation). Further addition of water

to this phase would decrease the oil fraction in the mixture to a point where the water cavities, formed inside the inverse micelles, transform into hexagonal arrays of cylindrical water channels (Lisiecki et al., 1999). Our week-long humidity experiments, which revealed this inverse hexagonal phase, suggest that the inverse micellar phase observed in these levitated particles is a transient phase on its way to becoming the inverse hexagonal phase observed under the polarising microscope – consistent with the excess water phase observed in bulk mixtures with water of the same organic composition (Fig. S9 and

S12, which includes a cartoon of the inverse hexagonal phase). These phases are known to have differing physical characteristics such as viscosities (Mezzenga et al., 2005; Tiddy, 1980), which have a significant effect on diffusion through the particle phase, affecting reactive gas and water uptake and the rate of these processes (Koop et al., 2011; Marshall et al., 2016; Mikhailov et al., 2009; Reid et al., 2018; Shiraiwa et al., 2011).

There are implications for the atmospheric lifetime of such particles in locations where oleic acid is likely to be

found such as in cooking emissions in urban areas of the UK (Allan et al., 2010) and China (Zhao et al., 2015), even emissions from a university cafeteria (Alves et al., 2020). We have shown that the dry acid–soap complex is much more stable towards ozonolysis than liquid oleic acid (Fig. 2(i)), having an ~ 80 % lower reactivity. If humidity-dependent phase changes take *ca.* a few hours to occur (~ 4 h for the large particle studied here), viscosity and diffusion through the particle would also change on a similar timescale. Reactive uptake of oxidants such as ozone would therefore evolve slowly,

resulting in a varying particle lifetime in the atmosphere that is dependent on its surrounding humidity - one of the uncertainties that need to be considered in atmospheric models (Abbatt et al., 2012; Gallimore et al., 2011; Liao et al., 2004).

Ozonolysis eventually breaks down the acid–soap complex. The reaction was stopped after 6+ h (400 min) exposure to a high ozone concentration (Fig. 2(i)). 34 ± 8 % of the double bonds remain in the particle as demonstrated by simultaneous Raman microscopy, while the acid–soap structure was no longer observable by SAXS from ~ 300 min onwards

with 59 ± 8 % of the double bonds still remaining at that point. Atmospheric ageing thus changes the relative amounts of oleic acid and sodium oleate in the particle, limiting the ability of the acid–soap to remain stable and slowly breaking down





the structure. The fact that the reaction remains slow after the acid–soap complex has disappeared indicates that the diffusivity of oleic acid and ozone continues to be severely impeded, thus maintaining the low reaction rate.

The aged particle is a more complicated mixture of reactants and products, some of which could be surface active
and affect the phases observed within the particle. The presence of surface-active organic material in atmospheric aerosols is known to affect aerosol hygroscopicity by affecting properties such as the surface tension of aqueous droplets (Bzdek et al., 2020; Facchini et al., 1999, 2000; Ovadnevaite et al., 2017). Two of the primary products of oleic acid ozonolysis have been reported to be surface active: azelaic acid (Tuckermann, 2007) and nonanoic acid (King et al., 2009). If a significant portion of oleic acid remains in the particle as we have reported, it is likely that some of these surface-active products also remain in
the viscous particle, although formation of high molecular weight products derived from these molecules would act as a sink (Zahardis et al., 2006). This implies that in addition to the preservation of oleic acid in the particle, other surface-active products could also be preserved. Increasing the residence time of these surfactant molecules could have significant effects on cloud formation (Facchini et al., 1999; Prisle et al., 2012). These findings are also consistent with organic residues observed after oxidation of unsaturated organic films at the air–water interface, with similar implications (King et al., 2009;
Pfrang et al., 2014; Sebastiani et al., 2018; Woden et al., 2018, 2021).

Products from oleic acid ozonolysis are known to take part in oligomerisation reactions with Criegee intermediates (Lee et al., 2012; Reynolds et al., 2006; Wang et al., 2016; Zahardis et al., 2006). These high molecular weight species are likely to have different physical characteristics, *e.g.* lower bulk diffusion coefficients and higher viscosities, compared to their precursors and other components in the particle. The higher molecular weight products may therefore impede reactant
diffusion and act as a "crust", as proposed in previous modelling studies (Pfrang et al., 2011; Zhou et al., 2019). This is consistent with the continued slow ozonolysis after the acid–soap complex degradation we report here.

We suggest that there is a phase separation between viscous products and unreacted oleic acid at the end of the ozonolysis experiment, as suggested in the schematic Fig. 3. This separation is in the form of a core–shell arrangement where the shell consists of a majority high-molecular-weight, viscous product phase (evidenced by the low-$q$ scattering
observed in the oxidised particle, Fig. 2(g) and Fig. S8) and the core is a majority unreacted disordered oleic acid phase. This is opposite to the viscous core-less viscous shell behaviour observed during humidification. Unlike the core–shell behaviour observed during humidification, where both phases are observable by SAXS, the unreacted (and disordered) oleic acid is not detectable by SAXS. However, simultaneous WAXS has enabled us to determine that unreacted free oleic acid (with a characteristic WAXS peak, Fig. S11) does exist both in the core and the shell of the particle at the end of ozonolysis (Fig.
2(h)). The fact that: (i) the Raman laser was focussed on the bulk of the sample, showing only disordered oleic acid by the end of the reaction; and (ii) ozonolysis does not speed up after the acid–soap SAXS signal has gone means it is most likely that a core–shell morphology prevails and that the reaction becomes limited by this inert shell (Pfrang et al., 2011). This also fits with the previous suggestion that viscous and large particles are not well-mixed and that the reaction occurs primarily in the surface layers of the particle (Moise and Rudich, 2002). The boundary between these phases is not thought to be as





distinct as in the core–shell system observed during humidification due to the presence of oleic acid WAXS signal both in the centre and the edge of the particle.

The phase state of organic aerosols has been shown to affect the chemistry and transport of harmful pollutants, such as Polycyclic Aromatic Hydrocarbons (PAHs) (Mu et al., 2018). A coating of organic material on an aerosol particle has been used to explain the long-range atmospheric transport of toxic PAHs, which are products of combustion, increasing

lung cancer risks (Shrivastava et al., 2017). The authors stated that the shielding was viscosity-dependent, therefore depending on the on the phase of the organic layer. In this study, atmospheric ageing of the proxy (whether by humidity change or oxidation) significantly affects the phase of the particle by either changing the 3–D molecular arrangement or destroying the self-assembly altogether. Additionally, an inert crust layer formed as a result of oxidation may contribute to the shielding effect. As oleic acid has been observed in the urban environment, viscous oleic acid-derived phases (including

the acid–soap complex) could contribute to the effect organic films have on the shielding and transport of pollutants and thus on public health. Indeed, oleic acid has been observed in PAH-containing marine organic aerosols with significant urban influences, suggesting that this shielding effect could contribute to the transport of such aerosols (Kang et al., 2017).

Temperature has been identified as a key factor in determining phase state, reactivity and atmospheric transport of a reactive aerosol species (Mu et al., 2018). The acid–soap complex is stable at room temperature (~ 22 ℃). Our POM

temperature experiments also show that the acid–soap complex is thermally stable until degradation at ~ 32 ℃, i.e. in the atmospherically relevant range (Fig. S5) consistent with the literature (Tandon et al., 2001). This suggests that this crystalline phase could exist in warm and dry environments. Although temperature would affect key parameters such as reaction rate constants and reactive gas surface accommodation times, the observation of a crystalline phase of oleic acid up to ~ 32 ℃ implies that reactant diffusion arguments made here are valid in many atmospheric temperature conditions.

Heterogeneous oleic acid ozonolysis has been reported to be a surface reaction with a small reacto-diffusive length of ~ 10–20 nm (Mendez et al., 2014; Moise and Rudich, 2002; Morris et al., 2002; Smith et al., 2002). A study on a different solid (in this case frozen) form of oleic acid showed that the reactive uptake of ozone is significantly decreased due to its solidity - the reaction happens only at the surface and diffusion to and from the near-surface and bulk layers is severely impeded (Moise and Rudich, 2002). The same argument can be applied to the acid–soap complex, which is stable at

significantly higher temperatures than the frozen oleic acid investigated previously.

These observations emphasise the importance of the effect that solid/semi-solid species have on the viscosity and diffusion of reactants, and therefore oxidation kinetics. Atmospheric lifetimes of organic aerosols can be significantly increased as a result of viscous phase formation (Shiraiwa et al., 2010, 2011a; Virtanen et al., 2010). Here, in addition to previously-reported liquid crystalline phases (Pfrang et al., 2017), a solid crystalline state of an unsaturated fatty acid aerosol

proxy was oxidised and results clearly show that the crystalline nature of the particle is the reason for the retardation of the reaction rate, which we estimate as being ~ 80 % slower for the acid–soap complex compared to an oleic acid droplet of similar size.





We are now able to spatially resolve SAXS/WAXS patterns through an acoustically levitated particle during humidity changes, revealing structural and physical changes as a result. This has provided a droplet-level picture of a diffusion front forming in a humidifying particle whereby an inverse micellar phase starts to form on the outside of the particle before eventually forming the dominant particle phase. This is in line with literature observations of diffusion fronts in highly viscous aerosol particles (Bastelberger et al., 2018; Zobrist et al., 2011). If diffusion varies across a particle, it follows that the diffusion coefficients of atmospheric trace gases would also vary throughout the particle. Some gases, such as ozone, are more soluble in hydrophobic than hydrophilic solvents (Panich and Ershov, 2019). The inverse micellar phase is a "water-in-oil" phase – where pockets of water are enclosed by surfactant molecules with their hydrophobic chains forming the majority hydrophobic "oil" domain (see Fig. 3). This means that ozone uptake is expected to increase upon inverse micelle formation for two reasons: (i) the phase is less viscous than the solid acid–soap complex, increasing the rate of ozone dissolution; and (ii) the majority of the particle is hydrophobic (water-in-oil), suggesting ozone is more likely to dissolve and diffuse through the hydrophobic region of the particle rather than being constrained inside pockets of water.

## 5 Conclusion

The oleic acid/sodium oleate acid–soap complex has been identified in an unsaturated fatty acid aerosol proxy. Raman and IR spectroscopy, along with SAXS/WAXS, were used to confirm the formation of the acid–soap complex in acoustically levitated particles.

We observed a clear phase gradient in a humidifying levitated acid–soap complex consistent with impeded diffusion of water through highly viscous (in this case, solid and semi-solid) aerosols (Koop et al., 2011; Mikhailov et al., 2009; Pfrang et al., 2011; Reid et al., 2018). This is the first time that a spatially resolved phase gradient throughout a levitated humidifying aerosol particle has been reported using an X-ray-based technique. While we are unable to quantify the viscosity of the particle directly, it is possible to probe changes at the molecular organisational level, identifying clear differences in the way oleic acid moieties organise themselves during atmospheric ageing. Preceding literature has reported mapped viscosity changes across a humidifying and oxidising aerosol particle using a fluorescence-based technique: Fluorescence Lifetime Imaging Microscopy (FLIM) (Hosny et al., 2013, 2016), with some experiments on optically-levitated particles (Athanasiadis et al., 2016; Fitzgerald et al., 2016). An advantage of an X-ray-based technique in comparison to FLIM is that there is no need to add a molecular marker to the sample in order to measure physical changes. This is especially important when considering self-assembled systems as adding other molecules to the system is likely to change the self-assembled structure (Salentinig et al., 2010). The SAXS/WAXS experiment can also be used on samples in any physical state found in the atmosphere, allowing for phase changes to be monitored under a variety of conditions. Signals in the WAXS pattern correlate with crystallinity for this system and also contain information about the packing of alkyl chains via characteristic scattering peaks (Fig. 1(b) and Fig. S13). We have also shown that it is possible to observe product formation via low-$q$ scattering intensity increases, opening a new avenue of inquiry.


An aerosol particle will experience changes in humidity during its lifetime in the atmosphere. A gradual phase change throughout the humidifying acid–soap complex suggests that the particle may rarely be in a homogeneous state; rather, there would likely be humidity-dependent physical differences within the particle as its atmospheric environment is changing.

        Week-long exposure to high humidity revealed the inverse hexagonal liquid crystal phase observed under the
polarising microscope, correlating with the inverse hexagonal phase observed in bulk mixtures of oleic acid/sodium oleate with excess water. This is therefore believed to be the equilibrium phase at saturated humidity. Molecular diffusion through liquid crystal phases, such as the inverse hexagonal phase, can vary significantly and has been used for this reason with regard to drug delivery (Zabara and Mezzenga, 2014). Analogy can be drawn with the dissolution of atmospheric species whereby uptake of atmospheric trace gases could vary significantly depending on the 3–D organisation of the surfactant
molecules.

        The present study demonstrates that oleic acid will have a longer atmospheric lifetime if incorporated in an acid–soap complex. Aerosols have significant impacts on urban air pollution with organic aerosol emissions as key components (Chan and Yao, 2008). In the UK, cooking organic aerosols have been estimated as an additional 10 % of anthropogenic $PM_{2.5}$ emissions (Ots et al., 2016). As previously discussed, phase-dependent viscous organic coatings have been reported to
shield harmful carcinogenic combustion products (Shrivastava et al., 2017). Inert organic layer formation, such as what was observed in this study, may therefore contribute to this shielding effect, since a gas-phase oxidant cannot reach even highly reactive compounds in our particles.

        The modelling of organic atmospheric aerosols and their effect on the climate is of great importance to the climate community (Jimenez et al., 2009; Kanakidou et al., 2005). If an aerosol has an extended atmospheric lifetime, it is more
likely to affect the climate and urban environment. The physical state-dependent reactivity of the unsaturated fatty acid aerosol proxy presented here suggests that its atmospheric lifetime is variable and that this variance is significant. The addition of the humidity-dependent phase changes observed in the levitated proxy further enhances the dynamic nature of this system, highlighting how physical state is of utmost importance when considering reactivity and atmospheric lifetimes.

        In summary, we have shown that the acid–soap complex is formed in an unsaturated fatty acid atmospheric aerosol
proxy. This acid–soap complex is stable under atmospheric conditions and can be formed from an inverse micellar phase as demonstrated by reversible phase changes during a humidification–dehumidification cycle. A core–shell effect was observed during this cycle and phase-dependent diffusivity was estimated with ~ 33-fold difference between crystalline core and liquid crystalline shell. The proxy's ozone reactivity reduces significantly in the acid–soap complex state, and this remained so even after the acid–soap complex broke down. Ozonolysis does not go to completion after the reaction was allowed to
continue for ~ 6 h at a high ozone concentration: $34 \pm 8$ % of initial oleic acid remains in the particle and low-q scattering is observed in the SAXS pattern, suggesting that high-molecular-weight/oligomeric products are present and these products exhibit some order. This is evidence for an inert crust formation, inhibiting particle reactivity and protecting surface-active molecules from ageing with implications for aerosol processes such as cloud formation. This layer may also help protect



toxic aerosol components such as PAHs, enabling them to travel further. This study presents a novel way of obtaining a
spatially resolved phase picture of single aerosol particles, with the addition of WAXS to the list of simultaneous
experiments possible on an acoustically levitated particle. We continue to demonstrate the versatile nature of oleic acid as an
unsaturated fatty acid aerosol proxy.

*Data availability.* Data related to this study are available in the Supplement as a .zip file and a .pdf. Raw
SAXS/WAXS/Raman data are available from the corresponding author upon request.

*Author contributions.* AM carried out experiments, processed/analysed the data and co-wrote the manuscript; CP led the
design and development of the acoustic levitator, initiated and co-designed the research project, carried out experiments,
contributed to data analysis and co-wrote the manuscript; AMS co-designed the research project, carried out experiments,
contributed to data analysis and co-wrote the manuscript; NJT set up and provided support during experiments on the I22
beamline at the Diamond Light Source; ADW set up and coupled the Raman experiment with the acoustic levitator and
provided support during beamtime experiments. JB helped at beamtime experiments and contributed to sample preparation

*Acknowledgements.* AM wishes to thank NERC SCENARIO DTP award number NE/L002566/1 and CENTA DTP; CP
wishes to thank the Royal Society (2007/R2) and NERC (grants NE/G000883/1 and NE/G019231/1) for support to develop
the acoustic levitation system; JB was funded by the EPSRC Centre for Doctoral Training in Sustainable Chemical
Technologies EP/L016354/1; Staff on the I22 beamline at the Diamond Light Source including Andy Smith and Tim Snow
are acknowledged; Niclas Johansson and Esko Kokkonen are acknowledged for their help at beamtime experiments. Ben
Woden is acknowledged for helping to calibrate the ozoniser. This work was carried out with the support of the Diamond
Light Source, instrument I22 (proposals SM20541 and SM21663). Joanne M. Elliott is acknowledged for providing access to
the polarising microscope.

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

Contributions from transport , solid fuel burning and cooking to primary organic aerosols in two UK cities, Atmos. Chem.
Phys., 10, 647–668, 2010.





Alves, C. A., Vicente, E. D., Evtyugina, M., Vicente, A. M., Nunes, T., Lucarelli, F., Calzolai, G., Nava, S., Calvo, A. I., Alegre, C. del B., Oduber, F., Castro, A. and Fraile, R.: Indoor and outdoor air quality: A university cafeteria as a case study, Atmos. Pollut. Res., 11(3), 531–544, doi:10.1016/j.apr.2019.12.002, 2020.

Ananthapadmanabhan, K. P. and Somasundaran, P.: Acid-soap formation in aqueous oleate solutions, J. Colloid Interface Sci., 122(1), 104–109, doi:10.1016/0021-9797(88)90293-7, 1988.

Athanasiadis, A., Fitzgerald, C., Davidson, N. M., Giorio, C., Botchway, S. W., Ward, A. D., Kalberer, M., Pope, F. D. and Kuimova, M. K.: Dynamic viscosity mapping of the oxidation of squalene aerosol particles, Phys. Chem. Chem. Phys., 18(44), 30385–30393, doi:10.1039/c6cp05674a, 2016.

Bastelberger, S., Krieger, U. K., Luo, B. P. and Peter, T.: Time evolution of steep diffusion fronts in highly viscous aerosol particles measured with Mie resonance spectroscopy, J. Chem. Phys., 149(24), 244506, doi:10.1063/1.5052216, 2018.

Berkemeier, T., Steimer, S. S., Krieger, U. K., Peter, T., Pöschl, U., Ammann, M. and Shiraiwa, M.: Ozone uptake on glassy, semi-solid and liquid organic matter and the role of reactive oxygen intermediates in atmospheric aerosol chemistry, Phys. Chem. Chem. Phys., 18(18), 12662–12674, doi:10.1039/c6cp00634e, 2016.

Blöchliger, E., Blocher, M., Walde, P. and Luisi, P. L.: Matrix Effect in the Size Distribution of Fatty Acid Vesicles, J. Phys. Chem. B, 102(50), 10383–10390, doi:10.1021/jp981234w, 1998.

Boucher, O., Randall, D., Artaxo, P., Bretherton, C., Feingold, G., Forster, P., Kerminen, V.-M., Kondo, Y., Liao, H., Lohmann, U., Rasch, P., Satheesh, S. K., Sherwood, S., Stevens, B. and Zhang, X. Y.: Clouds and Aerosols, in Climate Change 2013 the Physical Science Basis: Working Group I Contribution to the Fifth Assessment Report of the Intergovernmental Panel on Climate Change, edited by T. F. Stocker, D. Qin, G.-K. Plattner, M. Tignor, S. K. Allen, J. Boschung, A. Nauels, Y. Xia, V. Bex, and P. M. Midgley, pp. 571–658, Cambridge University Press, Cambridge, United Kingdom and New York, NY, USA., 2013.

Bzdek, B. R., Reid, J. P., Malila, J. and Prisle, N. L.: The surface tension of surfactant-containing, finite volume droplets, Proc. Natl. Acad. Sci. U. S. A., 117(15), 8335–8343, doi:10.1073/pnas.1915660117, 2020.

Chan, C. K. and Yao, X.: Air pollution in mega cities in China, Atmos. Environ., 42(1), 1–42, doi:10.1016/j.atmosenv.2007.09.003, 2008.

Cheng, Y., Li, S. M., Leithead, A., Brickell, P. C. and Leaitch, W. R.: Characterizations of cis-pinonic acid and n-fatty acids on fine aerosols in the Lower Fraser Valley during Pacific 2001 Air Quality Study, Atmos. Environ., 38(34), 5789–5800, doi:10.1016/j.atmosenv.2004.01.051, 2004.

Chu, S. N., Sands, S., Tomasik, M. R., Lee, P. S. and McNeill, V. F.: Ozone oxidation of surface-adsorbed polycyclic aromatic hydrocarbons: Role of PAH-surface interaction, J. Am. Chem. Soc., 132(45), 15968–15975, doi:10.1021/ja1014772, 2010.

Cistola, D. P., Atkinson, D., Hamilton, J. A. and Small, D. M.: Phase Behavior and Bilayer Properties of Fatty Acids: Hydrated 1:1 Acid-Soaps, Biochemistry, 25(10), 2804–2812, doi:10.1021/bi00358a011, 1986.



Davies, J. F. and Wilson, K. R.: Raman Spectroscopy of Isotopic Water Diffusion in Ultraviscous, Glassy, and Gel States in

Aerosol by Use of Optical Tweezers, Anal. Chem., 88(4), 2361–2366, doi:10.1021/acs.analchem.5b04315, 2016.

Engblom, J., Engström, S. and Fontell, K.: The effect of the skin penetration enhancer Azone® on fatty acid-sodium soap-water mixtures, J. Control. Release, 33(2), 299–305, doi:10.1016/0168-3659(94)00105-4, 1995.

Estillore, A. D., Trueblood, J. V. and Grassian, V. H.: Atmospheric chemistry of bioaerosols: Heterogeneous and multiphase reactions with atmospheric oxidants and other trace gases, Chem. Sci., 7(11), 6604–6616, doi:10.1039/c6sc02353c, 2016.

Facchini, M. C., Mircea, M., Fuzzi, S. and Charlson, R. J.: Cloud albedo enhancement by surface-active organic solutes in growing droplets, Nature, 401(6750), 257–259, doi:10.1038/45758, 1999.

Facchini, M. C., Decesari, S., Mircea, M., Fuzzi, S. and Loglio, G.: Surface tension of atmospheric wet aerosol and cloud/fog droplets in relation to their organic carbon content and chemical composition, Atmos. Environ., 34(28), 4853–4857, doi:10.1016/S1352-2310(00)00237-5, 2000.

Fitzgerald, C., Hosny, N. A., Tong, H., Seville, P. C., Gallimore, P. J., Davidson, N. M., Athanasiadis, A., Botchway, S. W., Ward, A. D., Kalberer, M., Kuimova, M. K. and Pope, F. D.: Fluorescence lifetime imaging of optically levitated aerosol: A technique to quantitatively map the viscosity of suspended aerosol particles, Phys. Chem. Chem. Phys., 18(31), 21710–21719, doi:10.1039/c6cp03674k, 2016.

Freedman, M. A.: Phase separation in organic aerosol, Chem. Soc. Rev., 46(24), 7694–7705, doi:10.1039/c6cs00783j, 2017.

Freedman, M. A.: Liquid–Liquid Phase Separation in Supermicrometer and Submicrometer Aerosol Particles, Acc. Chem. Res., 53(6), 1102–1110, doi:10.1021/acs.accounts.0c00093, 2020.

Fu, P. Q., Kawamura, K., Chen, J., Charrière, B. and Sempéré, R.: Organic molecular composition of marine aerosols over the Arctic Ocean in summer: Contributions of primary emission and secondary aerosol formation, Biogeosciences, 10(2), 653–667, doi:10.5194/bg-10-653-2013, 2013.

Gallimore, P. J., Achakulwisut, P., Pope, F. D., Davies, J. F., Spring, D. R. and Kalberer, M.: Importance of relative humidity in the oxidative ageing of organic aerosols: Case study of the ozonolysis of maleic acid aerosol, Atmos. Chem. Phys., 11(23), 12181–12195, doi:10.5194/acp-11-12181-2011, 2011.

Gallimore, P. J., Griffiths, P. T., Pope, F. D., Reid, J. P. and Kalberer, M.: Comprehensive modeling study of ozonolysis of oleic acid aerosol based on real-time, online measurements of aerosol composition, J. Geophys. Res., 122(8), 4364–4377,

doi:10.1002/2016JD026221, 2017.

Guarnieri, M. and Balmes, J. R.: Outdoor air pollution and asthma, Lancet, 383(9928), 1581–1592, doi:10.1016/S0140-6736(14)60617-6, 2014.

He, X., Leng, C., Pang, S. and Zhang, Y.: Kinetics study of heterogeneous reactions of ozone with unsaturated fatty acid single droplets using micro-FTIR spectroscopy, RSC Adv., 7(6), 3204–3213, doi:10.1039/C6RA25255A, 2017.

Hosny, N. A., Fitzgerald, C., Tong, C., Kalberer, M., Kuimova, M. K. and Pope, F. D.: Fluorescent lifetime imaging of atmospheric aerosols: A direct probe of aerosol viscosity, Faraday Discuss., 165, 343–356, doi:10.1039/c3fd00041a, 2013.





Hosny, N. A., Fitzgerald, C., Vyšniauskas, A., Athanasiadis, A., Berkemeier, T., Uygur, N., Pöschl, U., Shiraiwa, M., Kalberer, M., Pope, F. D. and Kuimova, M. K.: Direct imaging of changes in aerosol particle viscosity upon hydration and chemical aging, Chem. Sci., 7(2), 1357–1367, doi:10.1039/c5sc02959g, 2016.

Hung, H. M., Katrib, Y. and Martin, S. T.: Products and mechanisms of the reaction of oleic acid with ozone and nitrate radical, J. Phys. Chem. A, 109(20), 4517–4530, doi:10.1021/jp0500900, 2005.

Ishimaru, M., Toyota, T., Takakura, K., Sugawara, T. and Sugawara, Y.: Helical Aggregate of Oleic Acid and Its Dynamics in Water at pH 8, Chem. Lett., 34(1), 46–47, doi:10.1246/cl.2005.46, 2005.

Iwahashi, M., Yamaguchi, Y., Kato, T., Horiuchi, T., Sakurai, I. and Suzuki, M.: Temperature dependence of molecular
conformation and liquid structure of cis-9-octadecenoic acid, J. Phys. Chem., 95(1), 445–451, doi:10.1021/j100154a078, 1991.

Jimenez, J. L., Canagaratna, M. R., Donahue, N. M., Prevot, A. S. H., Zhang, Q., Kroll, J. H., DeCarlo, P. F., Allan, J. D., Coe, H., Ng, N. L., Aiken, A. C., Docherty, K. S., Ulbrich, I. M., Grieshop, A. P., Robinson, A. L., Duplissy, J., Smith, J. D., Wilson, K. R., Lanz, V. A., Hueglin, C., Sun, Y. L., Tian, J., Laaksonen, A., Raatikainen, T., Rautiainen, J., Vaattovaara, P.,
Ehn, M., Kulmala, M., Tomlinson, J. M., Collins, D. R., Cubison, M. J., Dunlea, J., Huffman, J. A., Onasch, T. B., Alfarra, M. R., Williams, P. I., Bower, K., Kondo, Y., Schneider, J., Drewnick, F., Borrmann, S., Weimer, S., Demerjian, K., Salcedo, D., Cottrell, L., Griffin, R., Takami, A., Miyoshi, T., Hatakeyama, S., Shimono, A., Sun, J. Y., Zhang, Y. M., Dzepina, K., Kimmel, J. R., Sueper, D., Jayne, J. T., Herndon, S. C., Trimborn, A. M., Williams, L. R., Wood, E. C., Middlebrook, A. M., Kolb, C. E., Baltensperger, U. and Worsnop, D. R.: Evolution of Organic Aerosols in the Atmosphere,
Science (80-. )., 326(5959), 1525–1529, doi:10.1126/science.1180353, 2009.

Kanakidou, M., Seinfeld, J. H., Pandis, S. N., Barnes, I., Dentener, F. J., Facchini, M. C., Van Dingenen, R., Ervens, B., Nenes, A., Nielsen, C. J., Swietlicki, E., Putaud, J. P., Balkanski, Y., Fuzzi, S., Horth, J., Moortgat, G. K., Winterhalter, R., Myhre, C. E. L., Tsigaridis, K., Vignati, E., Stephanou, E. G. and Wilson, J.: Organic aerosol and global climate modelling: a review, Atmos. Chem. Phys., 5(4), 1053–1123, doi:10.5194/acp-5-1053-2005, 2005.

Kang, M., Yang, F., Ren, H., Zhao, W., Zhao, Y., Li, L., Yan, Y., Zhang, Y., Lai, S., Zhang, Y., Yang, Y., Wang, Z., Sun, Y. and Fu, P.: Influence of continental organic aerosols to the marine atmosphere over the East China Sea: Insights from lipids, PAHs and phthalates, Sci. Total Environ., 607–608, 339–350, doi:10.1016/j.scitotenv.2017.06.214, 2017.

Keene, W. C., Pszenny, A. A. P., Maben, J. R., Stevenson, E. and Wall, A.: Closure evaluation of size-resolved aerosol pH in the New England coastal atmosphere during summer, J. Geophys. Res. D Atmos., 109(23), 1–16,
doi:10.1029/2004JD004801, 2004.

King, M. D., Thompson, K. C. and Ward, A. D.: Laser tweezers raman study of optically trapped aerosol droplets of seawater and oleic acid reacting with ozone: Implications for cloud-droplet properties, J. Am. Chem. Soc., 126(51), 16710–16711, doi:10.1021/ja044717o, 2004.



King, M. D., Rennie, A. R., Thompson, K. C., Fisher, F. N., Dong, C. C., Thomas, R. K., Pfrang, C. and Hughes, A. V.:

Oxidation of oleic acid at the air-water interface and its potential effects on cloud critical supersaturations, Phys. Chem. Chem. Phys., 11(35), 7699–7707, doi:10.1039/b906517b, 2009.

King, M. D., Rennie, A. R., Pfrang, C., Hughes, A. V. and Thompson, K. C.: Interaction of nitrogen dioxide (NO2) with a monolayer of oleic acid at the air-water interface - A simple proxy for atmospheric aerosol, Atmos. Environ., 44(14), 1822–1825, doi:10.1016/j.atmosenv.2010.01.031, 2010.

Knopf, D. A., Anthony, L. M. and Bertram, A. K.: Reactive uptake of O3 by multicomponent and multiphase mixtures containing oleic acid, J. Phys. Chem. A, 109(25), 5579–5589, doi:10.1021/jp0512513, 2005.

Koop, T., Bookhold, J., Shiraiwa, M. and Pöschl, U.: Glass transition and phase state of organic compounds: Dependency on molecular properties and implications for secondary organic aerosols in the atmosphere, Phys. Chem. Chem. Phys., 13(43), 19238–19255, doi:10.1039/c1cp22617g, 2011.

Last, D. J., Nájera, J. J., Wamsley, R., Hilton, G., McGillen, M., Percival, C. J. and Horn, A. B.: Ozonolysis of organic compounds and mixtures in solution. Part I: Oleic, maleic, nonanoic and benzoic acids, Phys. Chem. Chem. Phys., 11(9), 1427–1440, doi:10.1039/b815425b, 2009.

Lee, J. W. L., Carrascón, V., Gallimore, P. J., Fuller, S. J., Björkegren, A., Spring, D. R., Pope, F. D. and Kalberer, M.: The effect of humidity on the ozonolysis of unsaturated compounds in aerosol particles, Phys. Chem. Chem. Phys., 14(22), 8023–

8031, doi:10.1039/c2cp24094g, 2012.

Liao, H., Seinfeld, J. H., Adams, P. J. and Mickley, L. J.: Global radiative forcing of coupled tropospheric ozone and aerosols in a unified general circulation model, J. Geophys. Res. Atmos., 109(16), 1–33, doi:10.1029/2003JD004456, 2004.

Lisiecki, I., André, P., Filankembo, A., Petit, C., Tanori, J., Gulik-Krzywicki, T., Ninham, B. W. and Pileni, M. P.: Mesostructured fluids. 1. Cu(AOT)2-H2O-isooctane in oil rich regions, J. Phys. Chem. B, 103(43), 9168–9175,

doi:10.1021/jp991242s, 1999.

Liu, P., Song, M., Zhao, T., Gunthe, S. S., Ham, S., He, Y., Qin, Y. M., Gong, Z., Amorim, J. C., Bertram, A. K. and Martin, S. T.: Resolving the mechanisms of hygroscopic growth and cloud condensation nuclei activity for organic particulate matter, Nat. Commun., 9(1), 4076, doi:10.1038/s41467-018-06622-2, 2018.

Lynch, M. L.: Acid-soaps, Curr. Opin. Colloid Interface Sci., 2(5), 495–500, doi:10.1016/S1359-0294(97)80097-0, 1997.

Lynch, M. L., Pan, Y. and Laughlin, R. G.: Spectroscopic and thermal characterization of 1:2 sodium soap/fatty acid acid-soap crystals, J. Phys. Chem., 100(1), 357–361, doi:10.1021/jp952124h, 1996.

Lynch, M. L., Wireko, F., Tarek, M. and Klein, M.: Intermolecular Interactions and the Structure of Fatty Acid−Soap Crystals, J. Phys. Chem. B, 105(2), 552–561, doi:10.1021/jp002602a, 2002.

Marshall, F. H., Miles, R. E. H., Song, Y., Ohm, P. B., Power, R. M., Reid, J. P. and Dutcher, C. S.: Diffusion and reactivity

in ultraviscous aerosol and the correlation with particle viscosity, Chem. Sci., 7, 1298–1308, doi:10.1039/c5sc03223g, 2016.

Mauersberger, K., Barnes, J., Hanson, D. and Morton, J.: Measurement of the ozone absorption cross-section at the 253.7 nm mercury line, Geophys. Res. Lett., 13(7), 671–673, doi:10.1029/GL013i007p00671, 1986.





Mele, S., Söderman, O., Ljusberg-Wahrén, H., Thuresson, K., Monduzzi, M. and Nylander, T.: Phase behavior in the biologically important oleic acid/sodium oleate/water system, Chem. Phys. Lipids, 211(September 2017), 30–36, doi:10.1016/j.chemphyslip.2017.11.017, 2018.

Mendez, M., Visez, N., Gosselin, S., Crenn, V., Riffault, V. and Petitprez, D.: Reactive and nonreactive ozone uptake during aging of oleic acid particles, J. Phys. Chem. A, 118(40), 9471–9481, doi:10.1021/jp503572c, 2014.

Mezzenga, R., Meyer, C., Servais, C., Romoscanu, A. I., Sagalowicz, L. and Hayward, R. C.: Shear rheology of lyotropic liquid crystals: A case study, Langmuir, 21(8), 3322–3333, doi:10.1021/la046964b, 2005.

Mikhailov, E., Vlasenko, S., Martin, S. T., Koop, T. and Pöschl, U.: Amorphous and crystalline aerosol particles interacting with water vapor: Conceptual framework and experimental evidence for restructuring, phase transitions and kinetic limitations, Atmos. Chem. Phys., 9(24), 9491–9522, doi:10.5194/acp-9-9491-2009, 2009.

Milsom, A., Squires, A. M., Woden, B., Terrill, N. J., Ward, A. D. and Pfrang, C.: The persistence of a proxy for cooking emissions in megacities: a kinetic study of the ozonolysis of self-assembled films by simultaneous small and wide angle X-ray scattering (SAXS/WAXS) and Raman microscopy, Faraday Discuss., (Accepted Manuscript), doi:10.1039/D0FD00088D, 2021.

Moise, T. and Rudich, Y.: Reactive uptake of ozone by aerosol-associated unsaturated fatty acids: Kinetics, mechanism, and products, J. Phys. Chem. A, 106(27), 6469–6476, doi:10.1021/jp025597e, 2002.

Morris, J. W., Davidovits, P., Jayne, J. T., Jimenez, J. L., Shi, Q., Kolb, C. E., Worsnop, D. R., Barney, W. S. and Cass, G.: Kinetics of submicron oleic acid aerosols with ozone: A novel aerosol mass spectrometric technique, Geophys. Res. Lett., 29(9), 71-1-71–4, doi:10.1029/2002gl014692, 2002.

Mu, Q., Shiraiwa, M., Octaviani, M., Ma, N., Ding, A., Su, H., Lammel, G., Pöschl, U. and Cheng, Y.: Temperature effect on phase state and reactivity controls atmospheric multiphase chemistry and transport of PAHs, Sci. Adv., 4(3), eaap7314, doi:10.1126/sciadv.aap7314, 2018.

Nájera, J. J., Percival, C. J. and Horn, A. B.: Infrared spectroscopic evidence for a heterogeneous reaction between ozone and sodium oleate at the gas-aerosol interface: Effect of relative humidity, Int. J. Chem. Kinet., 47(4), 277–288, doi:10.1002/kin.20907, 2015.

Nikiforidis, C. V., Gilbert, E. P. and Scholten, E.: Organogel formation via supramolecular assembly of oleic acid and sodium oleate, RSC Adv., 5(59), 47466–47475, doi:10.1039/c5ra05336f, 2015.

Osterroht, C.: Extraction of dissolved fatty acids from sea water, Fresenius. J. Anal. Chem., 345(12), 773–779, doi:10.1007/BF00323009, 1993.

Ots, R., Vieno, M., Allan, J. D., Reis, S., Nemitz, E., Young, D. E., Coe, H., Di Marco, C., Detournay, A., Mackenzie, I. A., Green, D. C. and Heal, M. R.: Model simulations of cooking organic aerosol (COA) over the UK using estimates of emissions based on measurements at two sites in London, Atmos. Chem. Phys., 16(21), 13773–13789, doi:10.5194/acp-16-13773-2016, 2016.



Ovadnevaite, J., Zuend, A., Laaksonen, A., Sanchez, K. J., Roberts, G., Ceburnis, D., Decesari, S., Rinaldi, M., Hodas, N., Facchini, M. C., Seinfeld, J. H. and O'Dowd, C.: Surface tension prevails over solute effect in organic-influenced cloud droplet activation, Nature, 546(7660), 637–641, doi:10.1038/nature22806, 2017.

Panich, N. M. and Ershov, B. G.: Solubility of Ozone in Organic Solvents, Russ. J. Gen. Chem., 89(2), 185–189, doi:10.1134/S1070363219020026, 2019.

Pfrang, C., Shiraiwa, M. and Pöschl, U.: Chemical ageing and transformation of diffusivity in semi-solid multi-component organic aerosol particles, Atmos. Chem. Phys., 11(14), 7343–7354, doi:10.5194/acp-11-7343-2011, 2011.

Pfrang, C., Sebastiani, F., Lucas, C. O. M., King, M. D., Hoare, I. D., Chang, D. and Campbell, R. A.: Ozonolysis of methyl oleate monolayers at the air-water interface: Oxidation kinetics, reaction products and atmospheric implications, Phys. Chem. Chem. Phys., 16(26), 13220–13228, doi:10.1039/c4cp00775a, 2014.

Pfrang, C., Rastogi, K., Cabrera-Martinez, E. R., Seddon, A. M., Dicko, C., Labrador, A., Plivelic, T. S., Cowieson, N. and Squires, A. M.: Complex three-dimensional self-assembly in proxies for atmospheric aerosols, Nat. Commun., 8(1), 1724, doi:10.1038/s41467-017-01918-1, 2017.

Pöschl, U.: Atmospheric aerosols: Composition, transformation, climate and health effects, Angew. Chemie Int. Ed., 44(46), 7520–7540, doi:10.1002/anie.200501122, 2005.

Price, H. C., Mattsson, J., Zhang, Y., Bertram, A. K., Davies, J. F., Grayson, J. W., Martin, S. T., O'Sullivan, D., Reid, J. P., Rickards, A. M. J. and Murray, B. J.: Water diffusion in atmospherically relevant α-pinene secondary organic material, Chem. Sci., 6(8), 4876–4883, doi:10.1039/c5sc00685f, 2015.

Prisle, N. L., Asmi, A., Topping, D., Partanen, A.-I., Romakkaniemi, S., Dal Maso, M., Kulmala, M., Laaksonen, A., Lehtinen, K. E. J., McFiggans, G. and Kokkola, H.: Surfactant effects in global simulations of cloud droplet activation, Geophys. Res. Lett., 39(5), n/a-n/a, doi:10.1029/2011GL050467, 2012.

Putnam, C. D., Hammel, M., Hura, G. L. and Tainer, J. A.: X-ray solution scattering (SAXS) combined with crystallography and computation: defining accurate macromolecular structures, conformations and assemblies in solution, Q. Rev. Biophys., 40(3), 191–285, doi:10.1017/s0033583507004635, 2007.

Reid, J. P., Bertram, A. K., Topping, D. O., Laskin, A., Martin, S. T., Petters, M. D., Pope, F. D. and Rovelli, G.: The viscosity of atmospherically relevant organic particles, Nat. Commun., 9(1), 1–14, doi:10.1038/s41467-018-03027-z, 2018.

Renbaum-Wolff, L., Grayson, J. W., Bateman, A. P., Kuwata, M., Sellier, M., Murray, B. J., Shilling, J. E., Martin, S. T. and Bertram, A. K.: Viscosity of a-pinene secondary organic material and implications for particle growth and reactivity, Proc. Natl. Acad. Sci., 110(20), 8014–8019, doi:10.1073/pnas.1219548110, 2013.

Reynolds, J. C., Last, D. J., McGillen, M., Nijs, A., Horn, A. B., Percival, C., Carpenter, L. J. and Lewis, A. C.: Structural analysis of oligomeric molecules formed from the reaction products of oleic acid ozonolysis, Environ. Sci. Technol., 40(21), 6674–6681, doi:10.1021/es060942p, 2006.

Salentinig, S., Sagalowicz, L. and Glatter, O.: Self-Assembled Structures and pKa Value of Oleic Acid in Systems of Biological Relevance, Langmuir, 26(14), 11670–11679, doi:10.1021/la101012a, 2010.



Schwier, A. N., Sareen, N., Lathem, T. L., Nenes, A. and McNeill, V. F.: Ozone oxidation of oleic acid surface films decreases aerosol cloud condensation nuclei activity, J. Geophys. Res. Atmos., 116(16), D16202, doi:10.1029/2010JD015520, 2011.

Sebastiani, F., Campbell, R. A., Rastogi, K. and Pfrang, C.: Nighttime oxidation of surfactants at the air-water interface: Effects of chain length, head group and saturation, Atmos. Chem. Phys., 18(5), 3249–3268, doi:10.5194/acp-18-3249-2018,
815   2018.

Seddon, A. M., Richardson, S. J., Rastogi, K., Plivelic, T. S., Squires, A. M. and Pfrang, C.: Control of Nanomaterial Self-Assembly in Ultrasonically Levitated Droplets, J. Phys. Chem. Lett., 7(7), 1341–1345, doi:10.1021/acs.jpclett.6b00449, 2016.

Seddon, J. M., Bartle, E. A. and Mingins, J.: Inverse cubic liquid-crystalline phases of phospholipids and related lyotropic
systems, J. Phys. Condens. Matter, 2, SA285–SA290, doi:10.1088/0953-8984/2/S/043, 1990.

Shiraiwa, M., Pfrang, C. and Pöschl, U.: Kinetic multi-layer model of aerosol surface and bulk chemistry (KM-SUB): The influence of interfacial transport and bulk diffusion on the oxidation of oleic acid by ozone, Atmos. Chem. Phys., 10, 3673–3691, doi:10.5194/acp-10-3673-2010, 2010.

Shiraiwa, M., Ammann, M., Koop, T. and Poschl, U.: Gas uptake and chemical aging of semisolid organic aerosol particles,
Proc. Natl. Acad. Sci. U. S. A., 108(27), 11003–11008, doi:10.1073/pnas.1103045108, 2011.

Shiraiwa, M., Li, Y., Tsimpidi, A. P., Karydis, V. A., Berkemeier, T., Pandis, S. N., Lelieveld, J., Koop, T. and Pöschl, U.: Global distribution of particle phase state in atmospheric secondary organic aerosols, Nat. Commun., 8, 1–7, doi:10.1038/ncomms15002, 2017.

Shrivastava, M., Lou, S., Zelenyuk, A., Easter, R. C., Corley, R. A., Thrall, B. D., Rasch, P. J., Fast, J. D., Simonich, S. L.
M., Shen, H. and Tao, S.: Global long-range transport and lung cancer risk from polycyclic aromatic hydrocarbons shielded by coatings of organic aerosol, Proc. Natl. Acad. Sci. U. S. A., 114(6), 1246–1251, doi:10.1073/pnas.1618475114, 2017.

Slade, J. H., Ault, A. P., Bui, A. T., Ditto, J. C., Lei, Z., Bondy, A. L., Olson, N. E., Cook, R. D., Desrochers, S. J., Harvey, R. M., Erickson, M. H., Wallace, H. W., Alvarez, S. L., Flynn, J. H., Boor, B. E., Petrucci, G. A., Gentner, D. R., Griffin, R. J. and Shepson, P. B.: Bouncier Particles at Night: Biogenic Secondary Organic Aerosol Chemistry and Sulfate Drive Diel
Variations in the Aerosol Phase in a Mixed Forest, Environ. Sci. Technol., 53(9), 4977–4987, doi:10.1021/acs.est.8b07319, 2019.

Smith, G. D., Woods, E., DeForest, C. L., Baer, T. and Miller, R. E.: Reactive uptake of ozone by oleic acid aerosol particles: Application of single-particle mass spectrometry to heterogeneous reaction kinetics, J. Phys. Chem. A, 106(35), 8085–8095, doi:10.1021/jp020527t, 2002.

Stevens, B. and Feingold, G.: Untangling aerosol effects on clouds and precipitation in a buffered system, Nature, 461(7264), 607–613, doi:10.1038/nature08281, 2009.

Tabazadeh, A.: Organic aggregate formation in aerosols and its impact on the physicochemical properties of atmospheric particles, Atmos. Environ., 39(30), 5472–5480, doi:10.1016/j.atmosenv.2005.05.045, 2005.



Tandon, P., Raudenkolb, S., Neubert, R. H. H., Rettig, W. and Wartewig, S.: X-ray diffraction and spectroscopic studies of
oleic acid-sodium oleate, Chem. Phys. Lipids, 109(1), 37–45, doi:10.1016/S0009-3084(00)00207-3, 2001.

Tiddy, G. J. T.: Surfactant-water liquid crystal phases, Phys. Rep., 57(1), 1–46, doi:10.1016/0370-1573(80)90041-1, 1980.

Tuckermann, R.: Surface tension of aqueous solutions of water-soluble organic and inorganic compounds, Atmos. Environ.,
41(29), 6265–6275, doi:10.1016/j.atmosenv.2007.03.051, 2007.

Veghte, D. P., Altaf, M. B. and Freedman, M. A.: Size dependence of the structure of organic aerosol, J. Am. Chem. Soc.,
135(43), 16046–16049, doi:10.1021/ja408903g, 2013.

Vesna, O., Sax, M., Kalberer, M., Gaschen, A. and Ammann, M.: Product study of oleic acid ozonolysis as function of
humidity, Atmos. Environ., 43(24), 3662–3669, doi:10.1016/j.atmosenv.2009.04.047, 2009.

Vicente, E. D., Vicente, A., Evtyugina, M., Carvalho, R., Tarelho, L. A. C., Oduber, F. I. and Alves, C.: Particulate and
gaseous emissions from charcoal combustion in barbecue grills, Fuel Process. Technol., 176(April), 296–306,
doi:10.1016/j.fuproc.2018.03.004, 2018.

Virtanen, A., Joutsensaari, J., Koop, T., Kannosto, J., Yli-Pirilä, P., Leskinen, J., Mäkelä, J. M., Holopainen, J. K., Pöschl,
U., Kulmala, M., Worsnop, D. R. and Laaksonen, A.: An amorphous solid state of biogenic secondary organic aerosol
particles, Nature, 467(7317), 824–827, doi:10.1038/nature09455, 2010.

Wang, M., Yao, L., Zheng, J., Wang, X., Chen, J., Yang, X., Worsnop, D. R., Donahue, N. M. and Wang, L.: Reactions of
Atmospheric Particulate Stabilized Criegee Intermediates Lead to High-Molecular-Weight Aerosol Components, Environ.
Sci. Technol., 50(11), 5702–5710, doi:10.1021/acs.est.6b02114, 2016.

Woden, B., Skoda, M., Hagreen, M. and Pfrang, C.: Night-Time Oxidation of a Monolayer Model for the Air–Water
Interface of Marine Aerosols—A Study by Simultaneous Neutron Reflectometry and in Situ Infra-Red Reflection
Absorption Spectroscopy (IRRAS), Atmosphere (Basel)., 9(12), 471, doi:10.3390/atmos9120471, 2018.

Woden, B., Skoda, M. W. A., Milsom, A., Gubb, C., Maestro, A., Tellam, J. and Pfrang, C.: Ozonolysis of fatty acid
monolayers at the air–water interface: organic films may persist at the surface of atmospheric aerosols, Atmos. Chem. Phys.,
21(2), 1325–1340, doi:10.5194/acp-21-1325-2021, 2021.

Zabara, A. and Mezzenga, R.: Controlling molecular transport and sustained drug release in lipid-based liquid crystalline
mesophases, J. Control. Release, 188, 31–43, doi:10.1016/j.jconrel.2014.05.052, 2014.

Zahardis, J. and Petrucci, G. A.: The oleic acid-ozone heterogeneous reaction system: Products, kinetics, secondary
chemistry, and atmospheric implications of a model system - A review, Atmos. Chem. Phys., 7(5), 1237–1274,
doi:10.5194/acp-7-1237-2007, 2007.

Zahardis, J., LaFranchi, B. W. and Petrucci, G. A.: Photoelectron resonance capture ionization-aerosol mass spectrometry of
the ozonolysis products of oleic acid particles: Direct measure of higher molecular weight oxygenates, J. Geophys. Res. D
Atmos., 110(8), 1–10, doi:10.1029/2004JD005336, 2005.





Zahardis, J., LaFranchi, B. W. and Petrucci, G. A.: Direct observation of polymerization in the oleic acid-ozone heterogeneous reaction system by photoelectron resonance capture ionization aerosol mass spectrometry, Atmos. Environ., 40(9), 1661–1670, doi:10.1016/j.atmosenv.2005.10.065, 2006.

Zhang, Q., Jimenez, J. L., Worsnop, D. R. and Canagaratna, M.: A case study of urban particle acidity and its influence on
secondary organic aerosol, Environ. Sci. Technol., 41(9), 3213–3219, doi:10.1021/es061812j, 2007.

Zhao, X., Hu, Q., Wang, X., Ding, X., He, Q., Zhang, Z., Shen, R., Lü, S., Liu, T., Fu, X. and Chen, L.: Composition profiles of organic aerosols from Chinese residential cooking: Case study in urban Guangzhou, south China, J. Atmos. Chem., 72(1), 1–18, doi:10.1007/s10874-015-9298-0, 2015.

Zhou, S., Hwang, B. C. H., Lakey, P. S. J., Zuend, A., Abbatt, J. P. D. and Shiraiwa, M.: Multiphase reactivity of polycyclic
aromatic hydrocarbons is driven by phase separation and diffusion limitations, Proc. Natl. Acad. Sci. U. S. A., 116(24), 11658–11663, doi:10.1073/pnas.1902517116, 2019.

Zobrist, B., Soonsin, V., Luo, B. P., Krieger, U. K., Marcolli, C., Peter, T. and Koop, T.: Ultra-slow water diffusion in aqueous sucrose glasses, Phys. Chem. Chem. Phys., 13(8), 3514–3526, doi:10.1039/c0cp01273d, 2011.
