# Peer review of "An organic crystalline state in ageing atmospheric aerosol proxies: spatially resolved structural changes in levitated fatty acid particles"

_Atmospheric Chemistry and Physics, 2021_

## Author Comment (AC1)

**Supplementary Information: An Organic Crystalline State in Ageing Atmospheric Aerosol Proxies: Investigation of Spatially Resolved Structural Changes in Fatty Acid Particles.**

Adam Milsom[1], Adam M. Squires[2], Jacob A. Boswell[2], Nicholas J. Terrill[3], Andrew D. Ward[4] and Christian Pfrang[1,5]

[1] University of Birmingham, School of Geography, Earth and Environmental Sciences, Edgbaston, B15 2TT, Birmingham, UK.
[2] University of Bath, Department of Chemistry, South Building, Soldier Down Ln, Claverton Down, BA2 7AX, Bath, UK.
[3] Diamond Light Source, Diamond House, Harwell Science and Innovation Campus, Fermi Ave, OX11 0DE, Didcot, UK.
[4] Central Laser Facility, Rutherford Appleton Laboratory, Harwell Campus, OX11 0QX, Didcot, UK.
[5] Department of Meteorology, University of Reading, Whiteknights, Earley Gate, RG6 6BB, Reading, UK.

**S1. Acid-Soap Characterisation**

| | Lamellar *d*-spacing / nm (uncertainty) | |
|---|---|---|
| | This work [21 °C] | Literature [7]*** |
| **OA:SO (1:1) Acid–Soap** | 4.5773 (0.0001)* | 4.61 (0.05) [5 °C](1) |
| **Sodium Oleate** | 4.35 (0.02)** | 4.51 (0.05) [15 °C](2) |
| **Oleic Acid** | Liquid at this temperature | 4.14 (0.05) [5 °C](3) |

Table S1. Comparison of measured lamellar *d*-spacings for the oleic acid–sodium oleate acid–soap complex with pure sodium oleate and oleic acid (*:levitated particle; **: capillary coating, ***: bulk sample; OA: oleic acid; SO: sodium oleate).

| | WAXS Spacings / nm (uncertainty) | | | | | | | |
|---|---|---|---|---|---|---|---|---|
| **This work (21 °C)** | 0.467 (0.001) | 0.455 (0.004) | 0.444 (0.006) | 0.407 (0.001) | 0.399 (0.001) | 0.378 (0.002) | 0.369 (0.001) | 0.363 (0.001) |
| **Tandon et al. (1) (5 °C)** | 0.470 | 0.462 | 0.452 | 0.404 | 0.396 | 0.376 | 0.368 | 0.362 |
| **Tandon et al.(1) (30 °C)** | 0.471 | 0.451 | 0.412 | 0.408 | 0.400 | 0.379 | 0.371 | 0.365 |

Table S2. WAXS spacings measuring the repeat distance between scattering planes in the hydrophobic tail.

Table S1 compares our measured *d*-spacing with that obtained by Tandon *et al.* for the same system, but in a bulk sample and at a lower temperature. This table also includes the spacings of oleic acid and sodium oleate at low temperatures.(2, 3) The value for the acid–soap complex is significantly different from the oleic acid and sodium oleate *d*-spacings. It is however within the error of the 4.61 ± 0.05 nm quoted by Tandon *et al.*(1) for a bulk sample. The substantial difference in uncertainty is due to the difference in techniques used: the present study utilised synchrotron radiation many times more intense than the X-rays from the laboratory-based powder diffraction instrument used in the literature. The number quoted in this study therefore has much better statistics associated with it.

The WAXS spacings measured in this study in general agree with the literature values (Table S2). Signals in the WAXS region suggest that a sample is crystalline, exhibiting shorter distance order in addition to the longer distance lamellar spacings. These characteristic spacings measure the sub-cell packing arrangements. Tandon *et al.* computed three sets of sub-cell parameters which accommodate the spectroscopically-deduced parallel fatty acid chain packing, all with $O_{||}$ sub-cell symmetry.(1)

The melting temperature for this acid–soap complex has been reported as ~ 32 °C.(1) Exploiting the birefringent property of the acid–soap complex, a POM experiment was carried out on the acid–soap sample used in this study along with oleic acid–sodium oleate mixtures of varying ratios. The decomposition of the acid–soap structure occurred in agreement with the literature at ~ 32 °C. Further details are provided in the Supplement sect. S2.

Oleic acid and sodium oleate have markedly different Raman spectra (Fig. S1). The packing of the alkyl chains in sodium oleate is more ordered than that of oleic acid. This is exhibited by the difference in peak profile in the C–H stretching region (~ 2840–3050 cm$^{-1}$).(1–3) The acid–soap complex has features similar to sodium oleate due to its crystallinity. However, the acid–soap complex spectrum is clearly distinct from those of its constituent species (see Fig. S1: panel (a) *vs.* (b)–(d)).

[Figure]

Figure S1. Raman spectra of (a) the acid–soap complex; (b) a bulk mixture of oleic acid:sodium oleate (1:1 wt) 30 wt % in water - the hexagonal LLC phase; (c) oleic acid and (d) sodium oleate. All samples deposited on microscope slides. C–H stretching region is enlarged and displayed as an inset. Key peaks associated with alkyl chain ordering are labelled: I = 2887 cm$^{-1}$, II & III = 2854 cm$^{-1}$ and IV = 2884 cm$^{-1}$.

Figure S1 shows a comparison of the acid–soap complex (panel (a)) and its components (panels (c) and (d)). The Raman spectrum for a bulk LLC hexagonal phase (confirmed by SAXS) of oleic acid:sodium oleate (1:1 wt) 30 wt % in H$_2$O is also presented for comparison (panel (b)). The strong peak at 2887 cm$^{-1}$ is characteristic of the ordered packing of the acid–soap complex alkyl chains and is similar to the findings of Tandon *et al*,(1) distinguishing it from its components. The oleic acid and liquid crystal spectra exhibit a stronger peak at 2854 cm$^{-1}$ with the peak originally at 2887 cm$^{-1}$ significantly weaker, up-shifted and broadened, arising from the disordered state of the alkyl chains in both of these systems. Additionally, there are sharp C–C stretching peaks between ~ 1050–1150 cm$^{-1}$ which suggest that there are multiple trans conformers in the hydrocarbon chain.(1) Raman spectra of levitated acid–soap complex particles exhibited peaks in similar positions, however these spectra were subject to very high background scattering (see Supplement sect. S4). Superimposed Raman spectra of the C–H stretching region of these components are presented in the Supplement (Fig. S2).

[Figure]

Figure S2. Superimposed Raman spectra of the 2800-3100 cm$^{-1}$ region exhibiting differences in peak positions.

The carboxylate C=O bond peak is strong in IR spectroscopy. The position and intensity of this peak eludes to the environment the C=O bond is found in. This is structure-dependent: both oleic acid and sodium oleate have carboxyl peaks; oleic acid has its peak at 1707 cm$^{-1}$, while sodium oleate has its peak shifted by nearly 150 cm$^{-1}$ to 1558 cm$^{-1}$ (see Fig. S3).

[Figure]

Figure S3. IR spectra of the acid–soap complex, oleic acid and sodium oleate. The CH$_2$ scissoring region is in the range of 1400-1500 cm$^{-1}$. The C=O bond peak region is in the range of 1500-1800 cm$^{-1}$.

The acid–soap complex exhibits a mixture of these two C=O/(–CO$_2$)$^-$-related peaks. The carboxyl signal has been shifted slightly higher to 1712 cm$^{-1}$ and is broader, consistent with the findings of Tandon *et al.*(1) The broadening and partial disappearance of the two signals

associated with oleic acid and sodium oleate suggest that the carboxylate groups are involved in hydrogen bonding, reported previously for similar complexes: sodium palmitate–palmitic acid(1:2)(4) and $C_4 - C_{24}$ (1:1) acid–soap complexes.(5)

The appearance of an un-split signal in the $CH_2$ scissoring region at 1469 cm$^{-1}$ suggests the parallel chain arrangement as opposed to a perpendicular arrangement found in other acid–soaps.(1, 4) This region of the acid–soap IR spectrum is significantly different to its components.

**S2. SAXS/WAXS of the Levitated Particle Centre during Humidity Changes**

[Figure]

Figure S4. Evolution of the 1–D SAXS and WAXS patterns at the centre of a levitated acid–soap complex particle during humidification to 90 % RH (a) & (b) and dehumidification from 90 % RH to ~ 38 % RH (c) & (d). Peak positions for both SAXS and WAXS patterns are analogous to those in Fig. 1. The sharp first lamellar peak is labelled as 'Lam' and the broad inverse micellar peak is labelled as 'Mic' in panel (a) for clarity.

**S3. Polarising Optical Microscopy (POM) – Thermal Decomposition of the Acid–Soap Complex**

The oleic acid-sodium oleate acid–soap complex is reported to have a thermal decomposition temperature of ~ 32 °C.(1) As this complex is birefringent, it is possible to view it using cross-polarised light. The crystals exhibit a bright pattern (Fig. S5). Using a heating stage, it was possible to heat the sample from room temperature to the decomposition temperature reported in the literature.

[Figure]

Figure S5. Polarising microscopy-temperature experiments of differing oleic acid:sodium oleate weight ratios deposited from ethanol onto microscope slides. OA = Oleic acid, SO = Sodium oleate. (a) 1: 4, (b) 1:1 (acid-soap complex) and (c) 2:1. RH ~ 50 %.

The acid–soap complex clearly starts to break down at ~ 32 °C. The pattern started to disappear rapidly once the temperature approached this value. No more birefringence was observed and the sample became 'dark'. This is a qualitative visual confirmation of the literature's observations obtained by Raman microscopy and X-ray diffraction. On inspection of the reported literature phase diagram, the acid-soap at this composition breaks down and forms an isotropic liquid.(1)

Two other oleic acid:sodium oleate ratios were made and tested in the same way, 2:1 and 1:4. Interestingly, the 2:1 ratio mixture became fluid at ~ 27 °C and lost its birefringence at ~ 30 °C. This is consistent with the phase diagram for the dry oleic acid/sodium oleate system. According to the reported phase diagram, this system becomes an isotropic liquid above the decomposition temperature and the decomposition temperature decreases as a function of the amount of oleic acid in the system.(1) The 1:4 system was heated to 68 °C. Some of the birefringence disappeared at ~ 32 °C. This is ascribed to the acid–soap complex in this system breaking down. The sample remained birefringent for the whole experiment, suggesting that only sodium oleate was left in the crystalline state as its melting point is > 200 °C.

**S4. Raman Spectra of the Levitated Acid-Soap Complex**

[Figure]

Figure S6. Raman spectra of the levitated acid-soap complex before and after ozonolysis for 400 min ([$O_3$] = 51.9 ± 0.5 ppm). The carbon-carbon double bond peak used to measure reaction kinetics is labelled.

Figure S6 presents the Raman spectrum before and after ozonolysis. Note that the plot is not adjusted in order to stack the spectra, meaning that there is a significantly high background signal at the beginning of the experiment compared with the end. This is thought to be due to the high background scattering encountered when measuring Raman spectra of solid crystalline material such as the acid-soap complex.

The C=C bond has decreased in intensity compared to the C-H peak observed at ~ 1442 cm$^{-1}$ indicating that oleic acid has reacted. However, there is still a clear signal remaining at the end of the experiment corresponding to 34.0 ± 8.5 % of oleic acid remaining in the particle (see main text Fig. 3). Note the change in profile of the region 2750-3050 cm$^{-1}$. This is evidence of the acid-soap complex breaking down and is presented and discussed in the *Evolution of the SAXS pattern during Ozonolysis* section of the main text.

**S5. Optical Images of a Levitated Sodium Oleate Particle - Humidification**

[Figure]

Figure S7. Two optical images of a levitated sodium oleate particle before humidification (a) and after humidification to > 90 % RH for ~ 3 h (b). Red scale bar represents 50 μm.

A particle of sodium oleate was levitated at ~ 50 % RH and exposed to > 90 % RH for a prolonged period (~ 3 h) in order to demonstrate the change in size and shape of a solid hygroscopic particle, such as sodium oleate, after water uptake. The deliquescence point of sodium oleate has been measured to be 88 ± 2 %.(6) There is a clear difference in size and shape between a dry and deliquesced particle.

**S6. Low-q SAXS Evidence for High-Molecular-Weight Product Formation during Ozonolysis**

[Figure]

Figure S8. SAXS patterns of a levitated acid-soap complex before and after ozonolysis compared with an empty-levitator background. There is a clear increase in low-q scattering as a result of ozonolysis. [$O_3$] = 52 ± 0.5 ppm.

Figure S8 clearly shows an increase in low-q SAXS signal and the appearance of a shoulder at ~ 0.07 Å$^{-1}$ as a result of ozonolysis. Low-q scattering signals are observed for species with

large repeat distances between equivalent scattering centres. Structures formed by polymeric molecules exhibit patterns at low q due to the larger size of those molecules compared to our fatty acid precursors.(7) As the particle is a complex mixture of products, it is not possible to discern much about the structure from this scattering curve because of the impure nature of the scattering phase. Better low-q resolution would be required to study structures with larger repeat distances than the self-assembled structures of smaller molecules (oleic acid-sodium oleate) we focus on in this study. This can be achieved by increasing the sample-to-detector distance of the SAXS setup – not practicable during a time-critical synchrotron beamtime experiment. Nevertheless, the presence of a structure with low-q scattering peak even after ozonolysis suggests that products themselves exhibit some ordering.

**S7. POM of the Humidified and Dehumidified Acid–Soap Complex**

Visual and spectroscopic evidence for a phase separation was obtained using POM (see section S3) and Raman microscopy; a summary of which is in Fig. S9. A film of the acid–soap complex was deposited on a microscope slide and initially allowed to dry over 6 days. A POM picture was taken at room RH (~ 50 %, Fig S9(a)). The acid–soap complex sample was birefringent with characteristic lamellar "streaks". The sample was then exposed to a saturated humidity chamber which was created by half-filling a small container with ultrapure water and suspending the microscope slide within it. The sample was left in the chamber for 7 days.

[Figure]

Figure S9. POM images of the oleic acid:sodium oleate (1:1 wt) acid–soap complex. (a) At ~ 50 % RH, showing a birefringent needle-like/streaky lamellar pattern. (b) Immediately after removal from the saturated humidity chamber. A "charcoal-like" pattern is observed, suggesting a hexagonal phase. (c) 5 h after removal from humidity chamber. Birefringent needles have returned along with a non-birefringent phase. Raman spectra of each portion presented both sides – A–S = Acid–Soap Complex Peak, CR = Cosmic Ray.

A POM picture was taken immediately after removal from the humidity chamber and exhibited a "charcoal" texture under the polarising microscope known to be the hexagonal phase texture for this system(8) (Fig. S9(b)), which disappeared within ~ 5 min of being in room RH (~ 50 %). This is evidence for an inverse hexagonal lyotropic liquid crystal phase which has been

shown to form in the *potassium* oleate variant of this acid–soap complex(9) and also for the oleic acid/sodium oleate/water/NaCl solution system.(8, 10–12) The fast disappearance of this texture suggests that this phase exists at high water content. Indeed, the hexagonal phase has been observed in our offline SAXS experiments using bulk mixtures of this fatty acid composition in excess water (Fig. S12). The hexagonal phase was not observed in the SAXS patterns for levitated acid–soap complex particles. The POM samples were allowed to equilibrate for a week at > 90 % RH as opposed to 340 min for the levitated particles. Longer experiments are not practicable during a synchrotron beamtime. Therefore the inverse hexagonal phase may indeed form in a levitated acid-soap complex particle if left to equilibrate over a period of days.

5 h after removal from the humidification chamber into room RH (~ 50 %), a phase separation is observed. Polarising microscopy pictures, in combination with Raman spectra of the birefringent and non-birefringent regions, confirm that there are two physically-distinct phases present after a 5 h equilibration time. It is suggested that the birefringent phase, now presented as needle-like structures, is the acid–soap complex. This is confirmed by the Raman spectrum taken of this region (Fig. S9(c)). The non-birefringent region is assumed to be the inverse micellar phase, possibly with an excess of oleic acid as there was an excess to begin with.

**S8 Water content determination and water uptake/loss model**

[Figure]

Figure S10. (a) & (b) SAXS patterns during humidification and dehumidification at the centre of the particle. Initial and final patterns presented to show the broad micellar peak centre (~ 0.20-0.26 Å⁻¹) change position and intensity relative to the lamellar peak. (c) normalised

micellar-lamellar peak ratio vs time humidifying; (d) inverse micellar *d*-spacing vs time humidifying and dehumidifying.

As described in the *Water Diffusion Gradient during Humidity Change* section of the main text, the micellar-lamellar peak area ratio and micellar *d*-spacing were chosen as measures of water content for humidification and dehumidification, respectively. Micellar-lamellar peak area ratio data were noisy, especially once the particle had taken up a relatively large amount of water at ~ 230 min (Fig. S10(c)). Experimental micellar-lamellar peak area ratios which were greater than the median of ratios from 230 min onwards (*i.e.* the average maximum micellar-lamellar peak area ratio, accounting for the occasional large fluctuation) were set as that median value (2315) and this was assumed to be the value at maximum water content – all values are normalised to this number in Fig. S10(c). Very few datapoints were above this value and they occurred towards the end of the experiment, where the particle was completely inverse micellar (Fig. 1(I) in the main text – where there is no lamellar peak and integration of the lamellar peak position range returned the area of the noise around the background, resulting in some large apparent micellar-lamellar peak ratios).

[Figure]

Figure S11. A schematic representation of the water uptake (a) and loss (b) model employed in this study.

Internal water diffusion was evolved in a Vignes-type fashion(*e.g.* Price *et al.*),(13) relating water diffusion and layer composition (Eq. S1).

$$k_{internal,k} = (k_{micellar})^{\alpha [H_2O]_k} (k_{lamellar})^{1-\alpha [H_2O]_k} \qquad \text{(Eq. S1)}$$

$k_{internal,k}$ is the rate of internal diffusion in layer k; $k_{micellar}$ and $k_{lamellar}$ are the rates of water diffusion in the inverse micellar and crystalline lamellar phases, respectively; α is an activity coefficient which is assumed to be 1, analogous to the assumption of Davies and Wilson.(14) $[H_2O]_k$ is the amount of water as a fraction of the maximum water content in layer k, which is assumed to be the equilibrium water content for the inverse micellar phase.

The water uptake model (Fig. S11(a)) is described by Eq. S2 and S3:

$$\frac{d[H_2O]_{surf}}{dt} = k_{in}([H_2O]_{max} - [H_2O]_{surf}) + k_{internal}([H_2O]_k - [H_2O]_{surf}) \qquad \text{(Eq. S2)}$$

$$\frac{d[H_2O]_k}{dt} = k_{internal,k}([H_2O]_{k+1} + [H_2O]_{k-1} - 2[H_2O]_k) \qquad 2 < k < n-1 \qquad \text{(Eq. S3)}$$

The water loss model (Fig. S11(b)) is described by Eq. S4 with internal diffusion described by Eq. S3:

$$\frac{d[H_2O]_{surf}}{dt} = -k_{out}([H_2O]_{surf} - [H_2O]_{min}) + k_{internal}([H_2O]_k - [H_2O]_{surf}) \quad \text{(Eq. S4)}$$

The model splits the particle into a number of layers equivalent to the number of experimental positions ($n$) measured. The maximum ($[H_2O]_{max}$) and minimum ($[H_2O]_{min}$) amounts of water were set to 1 and 0, respectively in order to fit with the normalised experimental data. Each layer is given a number ($k$) with $k$ = 1 & n being the top and bottom surface layers, with their respective amounts of water defined as $[H_2O]_{surf}$ in Eq. S2 & S4. The key parameters varied to fit the model with the data were the rate of water uptake ($k_{in}$ – water uptake), the rate of water loss ($k_{out}$– water loss), $k_{micellar}$ and $k_{lamellar}$.

This model assumes: (i) constant rate of water uptake/loss into the particle – water uptake is expected to change with changing particle phase, however to avoid adding too many unknown parameters to the model we assume that it does not change; (ii) The particle is relatively flat and non-spherical – the particles levitated in this study are not spherical and spatially resolved data were of a vertical slice of the particle, the model reproduces this; (iii) there is a negligible rate of water loss during water uptake and *vice versa*; (iv) each model layer is well-mixed with no diffusion/water content gradient. Detailed modelling of differences in water uptake/loss rates into and from particles of different self-assembled phases is beyond the scope of this study and is the subject of ongoing work. The model presented here allows us to estimate the difference in water diffusivity between the inverse micellar and lamellar phases.

A Residual Sum of Squares (RSS) was calculated between the model and experiment and was used as a measure of goodness of fit, with lower values corresponding to better fits. Parameters were varied using a differential evolution algorithm whereby bounds are set for each parameter and parameter values are randomly selected from a population.(15) Each parameter is then "mutated" in an iterative process, each time the better-fitting parameter is kept. The algorithm eventually converges to an output which returns the minimum RSS value. The best fitting parameters are summarised in Table S3.

| Model | $k_{in}$ / x10$^{-3}$ | $k_{out}$ / x10$^{-3}$ | $k_{micellar}$ / x 10$^{-3}$ | $k_{lamellar}$ / x 10$^{-3}$ | $RSS_{fit}$ |
|---|---|---|---|---|---|
| Uptake | 3.9 | N/A | 590 | 16 | 209 |
| Loss | N/A | 23 | 600 | 18 | 25 |

Table S3. Optimised water uptake and loss model parameters with the minimised RSS ($RSS_{fit}$) quoted for both models.

Although the parameters obtained from the model have no physically meaningful units, the ratio between $k_{micellar}$ and $k_{lamellar}$ is ~ 33, highlighting the large difference in water diffusivity between the inverse micellar and crystalline lamellar phase. We must stress that the model fails to capture the prompt deliquescence well during humidification.

**S9. Hexagonal Phase in an Excess-Water Mixture of Oleic Acid/Sodium Oleate**

[Figure]

Figure S12. 1–D SAXS pattern of oleic acid-sodium oleate (1:1 wt) mixed with water as a 30 wt % organic mixture. Numbers above the peaks represent the characteristic peak position ratios expected for the inverse hexagonal phase. A cartoon representation of the inverse hexagonal phase is also presented.

It is known that mixtures of this composition make inverse hexagonal arrays of oleic acid-sodium oleate cylinders.(11) This phase can also be envisaged as a hexagonal array of water channels. This is the phase that produces the charcoal texture observed in the POM experiment (Fig. S9(b)). It is likely that this exists if the acid–soap complex is humidified for a time longer than the scope of a synchrotron experiment.

**S10. WAXS Pattern of Levitated Oleic Acid**

[Figure]

Figure S13. WAXS pattern of the centre of a levitated droplet of oleic acid compared with a background pattern of an empty levitator. This clearly demonstrates that oleic acid has a WAXS pattern. The alkyl chain spacing measured from this pattern is ~ 4.57 Å.

**References**

1.  P. Tandon, S. Raudenkolb, R. H. H. Neubert, W. Rettig, S. Wartewig, X-ray diffraction and spectroscopic studies of oleic acid-sodium oleate. *Chem. Phys. Lipids* **109**, 37–45 (2001).

2.  P. Tandon, R. Neubert, S. Wartewig, Thermotropic phase behaviour of sodium oleate as studied by FT-Raman spectroscopy and X-ray diffraction. *J. Mol. Struct.* **526**, 49–57 (2000).

3.  P. Tandon, G. Förster, R. Neubert, S. Wartewig, Phase transitions in oleic acid as studied by X-ray diffraction and FT- Raman spectroscopy. *J. Mol. Struct.* **524**, 201–215 (2000).

4.  M. L. Lynch, Y. Pan, R. G. Laughlin, Spectroscopic and thermal characterization of 1:2 sodium soap/fatty acid acid-soap crystals. *J. Phys. Chem.* **100**, 357–361 (1996).

5.  H. H. Mantsch, S. F. Weng, P. W. Yang, H. H. Eysel, Structure and thermotropic phase behavior of sodium and potassium carboxylate ionomers. *J. Mol. Struct.* **324**, 133–141 (1994).

6.  J. J. Nájera, Phase transition behaviour of sodium oleate aerosol particles. *Atmos. Environ.* **41**, 1041–1052 (2007).

7.    N. A. K. Meznarich, K. A. Juggernauth, K. M. Batzli, B. J. Love, Structural changes in PEO-PPO-PEO gels induced by methylparaben and dexamethasone observed using time-resolved SAXS. *Macromolecules* **44**, 7792–7798 (2011).

8.    S. Mele, *et al.*, Phase behavior in the biologically important oleic acid/sodium oleate/water system. *Chem. Phys. Lipids* **211**, 30–36 (2018).

9.    D. P. Cistola, D. Atkinson, J. A. Hamilton, D. M. Small, Phase Behavior and Bilayer Properties of Fatty Acids: Hydrated 1:1 Acid-Soaps. *Biochemistry* **25**, 2804–2812 (1986).

10.   J. M. Seddon, E. A. Bartle, J. Mingins, Inverse cubic liquid-crystalline phases of phospholipids and related lyotropic systems. *J. Phys. Condens. Matter* **2**, SA285–SA290 (1990).

11.   J. Engblom, S. Engström, K. Fontell, The effect of the skin penetration enhancer Azone® on fatty acid-sodium soap-water mixtures. *J. Control. Release* **33**, 299–305 (1995).

12.   C. Pfrang, *et al.*, Complex three-dimensional self-assembly in proxies for atmospheric aerosols. *Nat. Commun.* **8**, 1724 (2017).

13.   H. C. Price, *et al.*, Water diffusion in atmospherically relevant α-pinene secondary organic material. *Chem. Sci.* **6**, 4876–4883 (2015).

14.   J. F. Davies, K. R. Wilson, Raman Spectroscopy of Isotopic Water Diffusion in Ultraviscous, Glassy, and Gel States in Aerosol by Use of Optical Tweezers. *Anal. Chem.* **88**, 2361–2366 (2016).

15.   M. Wormington, C. Panaccione, K. M. Matney, D. K. Bowen, Characterization of structures from X-ray scattering data using genetic algorithms. *Philos. Trans. R. Soc. London A A* **357**, 2827–2848 (1999).

---

## Author Response (AR1)

We thank both of the reviewers for their comments and valuable time reviewing this manuscript. There seems to have been an oversight regarding the supporting information. We believed that a PDF with figures and descriptions was uploaded at submission with the .zip file containing the raw data (please note all of the references to the Supplement *e.g.* "Fig. S6" in the main text). This was accessible when the authors originally submitted the paper (and apparently available to reviewer 1). However, this PDF file seems not to be present in the link to the Supplement. Reviewer 2 therefore has raised justifiable concerns over the apparent lack of a supporting information document. We have included this PDF document with the revised manuscript (and made it available immediately via an author comment when reviewer 2 raised this) and ask that the editor check both the PDF and .zip file are included before any final decision is made. Most of the points raised by reviewer 2 are answered, at least in part, by the Supporting Information document as outlined below.

**Reviewer 1**

*1) As a proxy for cooking aerosol and sea spray aerosol the authors used a 1:1 wt ratio of oleic acid and sodium oleate. I am wondering how well this proxy represents cooking aerosol, or sea spray aerosol, or mixtures thereof? I assume that particles containing a 1:1 ratio of oleic acid and sodium oleate (without other organics or salts) are not present in the atmosphere. Hence, I am wondering how to extrapolate the results in the current study to the atmosphere.*

We thank the reviewer for their comment regarding the relevance of this proxy to the atmosphere. We note that the mixture is a molar excess of oleic acid (see beginning of sect. 2 of the main text). The fact that this acid-soap complex still forms at a molar excess of oleic acid shows that the sample does not need to be at an exact 1:1 composition, and that there is a composition window where it can form, where the 1:1 molar complex coexists with the excess component.

Atmospheric particulate matter is not always well mixed and its composition can be rather heterogeneous (Laskin et al., 2019). Fatty acids, including oleic acid, have been characterised on the surface of marine aerosols (Kirpes et al., 2019; Tervahattu, 2002; Tervahattu et al., 2005). This therefore means that they can concentrate in a specific region of a particle, and we suggest that 1:1 acid soap complexes can exist in some regions, in co-existence with other sections of different compositions. However, these results could contribute to an explanation for the extended lifetime of oleic acid in the atmosphere compared with laboratory investigations (Robinson et al., 2006; Rudich et al., 2007).

Our results suggest that it is possible to form such a phase at room humidity (~ 50 % RH).

The addition of other components to the mixture is the subject of upcoming publications which answer the reviewer's point that there are other molecules present in aerosol particles. The addition of other molecules does indeed change the phase observed, something explored qualitatively in previous work (Pfrang et al., 2017).

We have added a paragraph to the end of the atmospheric implications section (4), addressing this point and point **2** together (see next point).

*2) How sensitive are the result to the 1:1 ratio of oleic acid and sodium oleate? If different ratios are used or if other organic species or salts are added, do the authors expect the results to be completely different?*

As mentioned above, the composition does not need to be exactly at a 1:1 ratio, and there is a composition window overall where the 1:1 complex co-exists with the excess component. As pH is variable in the atmosphere (Paglione et al., 2021), so too would the ratio between oleic acid and sodium oleate. Beyond the composition window, we have data from an experiment on a 2:1 (oleic acid:sodium oleate) ratio particle, which we present below and will be added to the Supplement as Fig. S14. There is now a much larger excess of oleic acid in the particle and this resulted in an inverse micellar phase. Where exactly in the phase diagram this transition would occur is difficult to determine (small changes in composition are required) and are beyond the scope of this study. However, we can say that more oleic acid (more acidic conditions) induces the inverse micellar phase to form, consistent with the literature (Seddon et al., 1990). Again, follow-on work addresses the valid question of a more complex aerosol composition.

Regarding the addition of other molecules, although we have shown previously that they alter the lyotropic phases formed, they do not prevent lyotropic phase formation (Pfrang et al., 2017); we therefore suggest that the complex in the current work could still exist in the presence of other atmospheric components.

Two paragraphs in response to point **1** and **2** of reviewer 1 added to the *Atmospheric implications* section (4) are as follows:

[**Atmospheric particulate matter is not always well mixed and its composition can be rather heterogeneous (Laskin et al., 2019). Fatty acids, including oleic acid, have been characterised on the surface of marine aerosols** (Kirpes et al., 2019; Tervahattu, 2002; Tervahattu et al., 2005)**. This therefore means that they can concentrate in a specific region of a particle,** and we suggest that 1:1 acid-soap complexes can exist in some regions, in coexistence with other sections of different compositions**. The fact that this acid-soap complex still forms at a molar excess of oleic acid shows that the sample does not need to be at an exact 1:1 composition, and that there is a composition window where it can form,** where the 1:1 molar complex coexists with the excess component. **Our results suggest that it is possible to form such a phase at room humidity (~ 50 % RH). Formation of this phase could contribute to an explanation for the extended lifetime of oleic acid in the atmosphere compared with laboratory investigations (Robinson et al., 2006; Rudich et al., 2007).**

*As pH is variable in atmospheric aerosols (Paglione et al., 2021), so too would the ratio of oleic acid and sodium oleate. SAXS data from a 2:1 wt (oleic acid : sodium oleate) levitated particle, representing more acidic conditions, demonstrate that an inverse micellar phase forms at ~ 50 % RH (Fig. S14). Where exactly in the phase diagram this transition would occur is difficult to determine, requiring small changes in composition not practicable for a beamline experiment. This observation shows that a change in aerosol pH could affect particle viscosity via a change in nanostructure. The addition of other molecules does not prevent self-assembled phase formation and composition-dependent phase changes have been qualitatively observed (Pfrang et al., 2017) and are explored further in follow-on work.]*

[Figure]

[*Figure S14. A levitated particle of 2:1 wt (oleic acid : sodium oleate) composition demonstrating a broad peak characteristic of the inverse micellar phase with a d-spacing of 36 Å. RH ~ 50 %.*]

**Minor comment: The authors include changes due to relative humidity changes in the category of "aging". I would not refer to this as an aging process.**

We agree with the reviewer that this is not technically a direct ageing process. We have re-labelled section 3.2 as "**Atmospheric processing: (i) exposure to humidity changes**" and 3.3 as "**Atmospheric processing (ii): chemical ageing with ozone**". We have also amended references to "atmospheric ageing" to "atmospheric processing" when discussing humidification in section 4 (Atmospheric implications).

**Reviewer 2**

**1) In the Methodology, please provide a little more information on the experimental set up. Where was the ozone concentration measured and using what instrument/ports? For the POM analysis, how long were the samples allowed to equilibrate?**

The ozoniser was calibrated offline by UV-Vis spectroscopy as described in the manuscript. We did not make this as clear as we should have done and have now addressed this point directly by adding to section 2.3:

[*The ozone concentration was kept constant at 51.9 ± 0.5 ppm and was calibrated **offline** by UV/Vis spectroscopy **at the outlet of the ozoniser** using a **PerkinElmer Lambda 465 Spectrophotometer** and the ozone absorption band at 254 nm and the absorption cross-section for ozone at this wavelength (1.137 ± 0.070 x$10^{-17}$ cm$^2$)*]

The POM experiments are described in more detail in the Supporting Information (not originally accessible to reviewer 2). We have now included an additional description of the equilibration times in the main text in any case:

[*Samples **for humidity experiments were** deposited on microscope slides **and allowed to equilibrate at ~ 50 % RH over 6 days. Samples were then** humidified by suspending the slides above distilled water inside a small, sealed container. This provided a saturated environment for the samples to equilibrate with **for 7 days**. **Samples for temperature experiments were prepared in the same way but without any humidification.***]

**2) *For Figure 1, what was the RH value for each of the time points (g-l)?***

The experiment involved a step increase from ~38 to 90 % RH, with the target humidity being reached within 3 – 4 min. For the humidification plots (Fig. 1 (g) - (i)) only the first plot (g) was at 38 % RH. The rest were at 90 % RH. For the dehumidification plots (Fig. 1 (j) – (l)) again, the first plot was at 90 % RH and the rest at ~38 % RH (room RH). We apologise that this was not clear in the caption and this has now been amended accordingly:

[*Figure 1. ((a) and (b)) 1-D SAXS and WAXS patterns obtained from a dry levitated particle of the acid–soap complex – 1$^{st}$, 2$^{nd}$ and 3$^{rd}$ lamellar peaks are labelled and a cartoon of the lamellar phase is presented (a). ((c) and (d)) experimental fraction of maximum water content as a function of distance from particle centre and time humidifying/dehumidifying. ((e) and (f)) modelled fraction of maximum water content – best fits to experimental data for humidification and dehumidification. 3–D surface plots of 1–D SAXS patterns plotted against distance from the particle centre for the same particle humidifying ((g) – (i)) and dehumidifying ((j) – (l)) 180 with time humidifying/dehumidifying presented at the top right of each plot (particle size: ~ 150 µm (vertical radius) x 500 µm (horizontal radius); humidification experiment: ~ 38 % (room RH) **((g))** to 90 % **((h) and (i))** RH, dehumidification experiment: 90 % **(j)** to ~ 38 % (room RH) **((k) and (l))** RH).*]

**3) *Page 13, Figure 7 is mentioned but there is no Figure 7 in the manuscript.***

We thank the reviewer for spotting this typo carried over from an earlier figure arrangement. The figure referred to was Fig. 2(i). This has been corrected.

**4) In Section 3.3 a discussion is made about the ozonolysis experiments. However, no discussion of the role of ozonolysis products is included. Is it known if the particles loss mass? Is the levitator a closed system, or could semi-volatile products evaporate out of the particle? What effect might this have on the observed rates? The raw data for the C=C peak are not provided, please include these as a supplemental figure.**

We thank the reviewer for this comment. A brief description of the ozonolysis was provided at the beginning of section 3.3, however we understand the reviewer's point about not discussing the role of reaction products specifically. Our response is summarised in an additional paragraph at the beginning of section 3.3:

[*Of the reaction products, only nonanal is known to be volatile enough to evaporate appreciably (Vesna et al., 2009). The rest of the ozonolysis products are assumed to remain in the particle phase. There is evidence that particles of oleic acid lose a small proportion of mass during ozonolysis (~ 6 % mass loss after 20 h at 2 ppm), probably due to nonanal loss (Lee et al., 2012). If this is the case, a size change in these particles is likely to be smaller than can be resolved by the X-ray beam in these experiments (~ 15 µm in diameter). Being in an open system with a constant flow of oxygen and ozone, we cannot rule out any mass loss occurring during these experiments. Compared with the effect of particle phase state, we do not expect nonanal loss to impact significantly on the reaction rate.*]

We assume that the reviewer refers to the Raman spectra for the levitated particles. The Raman spectra before and after ozonolysis are included in the Supporting Information file.

**5) In section 3.3.3 it is stated that the weak shoulder at ~2854 cm-1 becomes more defined during oxidation and that this indicates the oleic acid left is not involved in the acid-soap structure. Please include what this shoulder corresponds to, how does the increase in definition show this?**

The shoulder at ~2854 cm$^{-1}$ corresponds to the –CH$_2$ symmetric stretching band in the free oleic acid, as opposed to the asymmetric stretching band of the ordered alkyl chains (~2887 cm$^{-1}$) (Tandon et al., 2001). There is a molar excess of oleic acid in the mixture, so this is to be expected. As ozonolysis proceeds this shoulder becomes a more defined peak. This area of the Raman spectrum now resembles that of liquid oleic acid. This spectrum is presented in the Supporting Information file along with a comparison with the other components of the mixture. We have amended the text accordingly to define these two bands more clearly.

[*Three key changes in the Raman spectrum are observed during ozonolysis. First, there is a clear shift of the strong acid–soap peak from ~ 2887 cm$^{-1}$ **(-CH$_2$ asymmetric stretching band)** to ~ 2897 cm$^{-1}$ accompanied by some broadening.*]

And:

[*Secondly, the weak shoulder at ~ 2854 cm$^{-1}$ **(-CH$_2$ symmetric stretch)** becomes a more defined **peak** during oxidation. **This region of the Raman spectrum resembles that of oleic acid (Fig. S1(c)).** This is further evidence, in combination with SAXS observations (Fig. 2(j)), that the oleic acid left in the system is not involved in an acid–soap structure after ozonolysis.*]

**6) Page 17 discusses inverse micellar vs. micellar. Please provide a little more information on why inverse micelles are expected for this system, even at the highest RH values.**

We thank the reviewer for this comment and understand the need for extra clarity on this point. We have added the following to the paragraph discussing this aspect in section 4:

 [... **The larger hydrophobic tail region of the oleic acid molecule compared with its hydrophilic head drives interface curvature towards water, and therefore the formation of inverse phases, even in excess water. Inverted micellar and other inverted topology phases have been observed for oleic acid – sodium oleate mixtures in excess water (Seddon et al., 1990).  Normal topology micelles (polar head groups at the micelle surface) form in systems with larger / charged headgroups, and are only observed within the sodium oleate – oleic acid system at high sodium oleate content (> 80 wt %) (Seddon et al., 1990). It is therefore likely that the micellar phase observed in this system has an inverse rather than normal topology suggested in the preceding atmospheric literature (see Fig. 3 for a cartoon representation).**]

**7) The data provided in the figures in the manuscript look like data for a single experiment. Were replicate experiments run for these samples? If so, which ones and how many replicates?**

The ozonolysis and humidity experiments were carried out in different particles. The vertical and horizontal radii are quoted in the figure captions for Fig. 1 and 2. We recognise that in the methods section 2.3 the final paragraph makes reference to the size of the particles but does not state the number studied. One particle was studied for the humidity experiment and three for the ozonolysis experiments (including the oleic acid particle) – see Fig. 2(i) in the main text.

The methods section 2.3 has been updated:

*[The levitated particles were analysed on the I22 beamline at the Diamond Light Source (UK). Solid samples, crystallised from ethanolic solutions, were placed into a node of the acoustic levitator. The particles had vertical radii of ~ 90–150 µm and horizontal radii of ~ 500 µm (determined using the attenuation of the X-ray beam). **One particle of proxy mixture was levitated subjected to the humidity change experiment. Two particles of the proxy mixture and one pure oleic acid particle were levitated and subjected to ozonolysis experiments.]***

The figure caption for Fig. 2 has also been updated for extra clarity:

[*Figure 2. Vertical scans through the particle showing the effect of ozonolysis on self-assembly. Each row of plots ((a) and (b), (c) and (d), (e) and (f), (g) and (h)) shows simultaneous 1–D SAXS and WAXS scattering patterns vs. distance from the particle centre (measured in µm from what was deemed the particle centre from attenuation data) at increasing time exposed to ozone (labelled at the top-right of every WAXS plot). The particle moved and possibly changed shape during the experiment, vertical movement is apparent from the SAXS and WAXS patterns. (i) Comparison of a levitated pure oleic acid droplet vs. a levitated acid–soap complex particle undergoing ozonolysis, measured by Raman microscopy - A longer ozonolysis experiment on a different levitated acid–soap complex particle is also presented, **totalling two ozonolysis experiments on this proxy**. (j) Evolution of the Raman*

*spectra between 2750 and 3050 cm$^{-1}$ of a levitated acid–soap complex during ozonolysis. (Particle size: ~ 85 µm (vertical radius) x ~ 500 µm (horizontal radius); [O3] = 51.9 ± 0.5 ppm).]*

An extra sentence has been added to the conclusion which clarifies that the acid-soap complex was also observed on microscope slide deposits:

[*The oleic acid/sodium oleate acid–soap complex has been identified in an unsaturated fatty acid aerosol proxy. Raman and IR spectroscopy, along with SAXS/WAXS, were used to confirm the formation of the acid–soap complex in acoustically levitated particles. **The acid–soap complex was also identified by Raman microscopy on microscope slide deposits**.*]

**Additional change:**

We noticed a typo in line 300 referring to a "Fig. 6". We have amended this to read "**Fig. 1(d) and (f)**".

**References**

Kirpes, R. M., Bonanno, D., May, N. W., Fraund, M., Barget, A. J., Moffet, R. C., Ault, A. P. and Pratt, K. A.: Wintertime Arctic Sea Spray Aerosol Composition Controlled by Sea Ice Lead Microbiology, ACS Cent. Sci., 5(11), 1760–1767, doi:10.1021/acscentsci.9b00541, 2019.

Laskin, A., Moffet, R. C. and Gilles, M. K.: Chemical Imaging of Atmospheric Particles, Acc. Chem. Res., 52(12), 3419–3431, doi:10.1021/acs.accounts.9b00396, 2019.

Lee, J. W. L., Carrascón, V., Gallimore, P. J., Fuller, S. J., Björkegren, A., Spring, D. R., Pope, F. D. and Kalberer, M.: The effect of humidity on the ozonolysis of unsaturated compounds in aerosol particles, Phys. Chem. Chem. Phys., 14(22), 8023–8031, doi:10.1039/c2cp24094g, 2012.

Paglione, M., Decesari, S., Rinaldi, M., Tarozzi, L., Manarini, F., Gilardoni, S., Facchini, M. C., Fuzzi, S., Bacco, D., Trentini, A., Pandis, S. N. and Nenes, A.: Historical Changes in Seasonal Aerosol Acidity in the Po Valley (Italy) as Inferred from Fog Water and Aerosol Measurements, Environ. Sci. Technol., acs.est.1c00651, doi:10.1021/acs.est.1c00651, 2021.

Pfrang, C., Rastogi, K., Cabrera-Martinez, E. R., Seddon, A. M., Dicko, C., Labrador, A., Plivelic, T. S., Cowieson, N. and Squires, A. M.: Complex three-dimensional self-assembly in proxies for atmospheric aerosols, Nat. Commun., 8(1), 1724, doi:10.1038/s41467-017-01918-1, 2017.

Robinson, A. L., Donahue, N. M. and Rogge, W. F.: Photochemical oxidation and changes in molecular composition of organic aerosol in the regional context, J. Geophys. Res. Atmos., 111(3), 1–15, doi:10.1029/2005JD006265, 2006.

Rudich, Y., Donahue, N. M. and Mentel, T. F.: Aging of Organic Aerosol: Bridging the Gap Between Laboratory and Field Studies, Annu. Rev. Phys. Chem., 58(1), 321–352, doi:10.1146/annurev.physchem.58.032806.104432, 2007.

Seddon, J. M., Bartle, E. A. and Mingins, J.: Inverse cubic liquid-crystalline phases of phospholipids and related lyotropic systems, J. Phys. Condens. Matter, 2, SA285–SA290, doi:10.1088/0953-8984/2/S/043, 1990.

Tandon, P., Raudenkolb, S., Neubert, R. H. H., Rettig, W. and Wartewig, S.: X-ray diffraction and spectroscopic studies of oleic acid-sodium oleate, Chem. Phys. Lipids, 109(1), 37–45, doi:10.1016/S0009-3084(00)00207-3, 2001.

Tervahattu, H.: Identification of an organic coating on marine aerosol particles by TOF-SIMS, J. Geophys. Res., 107(D16), 4319, doi:10.1029/2001JD001403, 2002.

Tervahattu, H., Juhanoja, J., Vaida, V., Tuck, A. F., Niemi, J. V., Kupiainen, K., Kulmala, M. and Vehkamäki, H.: Fatty acids on continental sulfate aerosol particles, J. Geophys. Res. D Atmos., 110(6), 1–9, doi:10.1029/2004JD005400, 2005.

Vesna, O., Sax, M., Kalberer, M., Gaschen, A. and Ammann, M.: Product study of oleic acid ozonolysis as function of humidity, Atmos. Environ., 43(24), 3662–3669, doi:10.1016/j.atmosenv.2009.04.047, 2009.

---

## Author Response (AR2)

We thank the reviewer for the comments on this revised manuscript. We note that points 1, 2, 3, 5, 6, 8 and 9 were all raised and specifically addressed in the previous review round, so we have left those responses unchanged from our previous author response. Point 4 and 7 are addressed specifically here and we have amended the Supporting Information accordingly.

1) In the Methodology, please provide a little more information on the experimental set up. Where was the ozone concentration measured and using what instrument/ports? For the POM analysis, how long were the samples allowed to equilibrate?

The ozoniser was calibrated offline by UV-Vis spectroscopy as described in the manuscript. We did not make this as clear as possible and have now addressed this point, adding to section 2.3:

[The ozone concentration was kept constant at 51.9  $\pm$  0.5 ppm and was calibrated **offline** by UV/Vis spectroscopy **at the outlet of the ozoniser** using a **PerkinElmer Lambda 465 Spectrophotometer** and the ozone absorption band at 254 nm and the absorption cross-section for ozone at this wavelength (1.137  $\pm$  0.070 x10-17 cm2)]

The POM experiments are described in more detail in the Supporting Information (not originally accessible to reviewer 2). We have now included an additional description of the equilibration times in the main text in any case:

[Samples for humidity experiments were deposited on microscope slides and allowed to equilibrate at ~ 50 % RH over 6 days. Samples were then humidified by suspending the slides above distilled water inside a small, sealed, container. This provided a saturated environment for the samples to equilibrate with for 7 days. Samples for temperature experiments were prepared in the same way but without any humidification.]

**2) For Figure 1, what was the RH value for each of the time points (g-I)?**

The experiment involved a step increase from ~38 to 90 % RH, with the target humidity being reached within 3 - 4 min. For the humidification plots (Fig. 1 (g) - (i)) only the first plot (g) was at 38 % RH. The rest were at 90 % RH. For the dehumidification plots (Fig. 1 (j) - (l)) again, the first plot was at 90 % RH and the rest at ~38 % RH (room RH). This is not clear in the caption and has been amended accordingly:

[Figure 1. ((a) and (b)) 1-D SAXS and WAXS patterns obtained from a dry levitated particle of the acid–soap complex – 1st, 2nd and 3rd lamellar peaks are labelled and a cartoon of the lamellar phase is presented (a). ((c) and (d)) experimental fraction of maximum water content as a function of distance from particle centre and time humidifying/dehumidifying. ((e) and (f)) modelled fraction of maximum water content – best fits to experimental data for humidification and dehumidification. 3–D surface plots of 1–D SAXS patterns plotted against distance from the particle centre for the same particle humidifying ((g) – (i)) and dehumidifying ((j) – (l)) 180 with time humidifying/dehumidifying presented at the top right of each plot (particle size: ~ 150 µm (vertical radius) x 500 µm (horizontal radius); humidification experiment: ~ 38 % (room RH) ((g)) to 90 % ((h) and (i)) RH, dehumidification experiment: 90 % (j) to ~ 38 % (room RH) ((k) and (l)) RH).]

**3) Page 13, Figure 7 is mentioned but there is no Figure 7 in the manuscript.**

We thank the reviewer for spotting this error. The figure referred to was Fig. 2(i). This has been amended.

**4) Images in Figure S9 were collected after the chemicals were deposited onto glass slides. What role does the substrate play in the observed structures in Figure S9? Can you provide scale bars to show the size of the crystals? Would these be relevant for aerosol particles?**

We believe that the substrate did not affect the structure formed on the glass slides. The Raman spectrum of the acid-soap complex exhibits the same strong peak at ~ 2887 cm-1 as demonstrated in the characterisation section (sect. S1) and in levitated particles of the same mixture (Fig. S4). The formation of the inverse hexagonal phase at high humidity is consistent with a bulk mixture of this organic composition with excess water (see sect. S9), where the sample vial would not have affected the resulting nanostructure. The non-birefringent phase observed after the humidification-dehumidification experiment is likely the inverse micellar phase, observed in levitated particles after humidity changes (see Fig. 1(I) in the main text and Fig. S4(c)). As we are confident that the glass substrate does not affect the observed structures, one can reasonably assume that this is the case for this non-birefringent phase.

This paragraph has been added to the Supporting Information sect. S7:

[We believe that the glass substrate did not affect the structure formed on the glass slides. The Raman spectrum of the acid-soap complex exhibits the same strong peak at ~ 2887 cm-1 as demonstrated in the characterisation section (sect. S1) and in levitated particles of the same mixture (Fig. S4). The formation of the inverse hexagonal phase at high humidity is consistent with a bulk mixture of this organic composition with excess water (see sect. S9), where the sample vial would not have affected the resulting nanostructure. The non-birefringent phase observed after the humidification-dehumidification experiment is likely the inverse micellar phase, observed in levitated particles after humidity changes (see Fig. 1(l) in the main text and Fig. S4(c)). As we are confident that the glass substrate does not affect the observed structures, one can reasonably assume that this is the case for this non-birefringent phase.]

Unfortunately, we are unable to provide scale bars for these microscope images. The purpose of these microscope slide experiments was to qualitatively observe nano-structural changes over a much longer time period than is practicable for a beamtime experiment. It is unlikely that large crystallites would form in real aerosol particles. See the response to reviewer 1 (point 2) in the previous review round where we discuss where acid-soap complexes could exist. Also see the additional discussion added to sect. 4 of the manuscript.

5) In Section 3.3 a discussion is made about the ozonolysis experiments. However, no discussion of the role of ozonolysis products is included. Is it known if the particles loss mass? Is the levitator a closed system, or could semi-volatile products evaporate out of the particle? What effect might this have on the observed rates? The raw data for the C=C peak are not provided, please include these as a supplemental figure.

We thank the reviewer for this comment. A brief description of the ozonolysis was provided at the beginning of section 3.3, however we understand the reviewer's point about not discussing the role of reaction products. Our response is summarised in an extra paragraph at the beginning of section 3.3:

[Of the reaction products, only nonanal is known to be volatile enough to evaporate appreciably (Vesna et al., 2009). The rest of the ozonolysis products are assumed to remain in the particle phase. There is evidence that particles of oleic acid lose a small proportion of mass during ozonolysis (~ 6 % mass loss after 20 h at 2 ppm), probably due to nonanal loss (Lee et al., 2012). If this is the case, a size change in these particles is likely to be smaller than can be resolved by the X-ray beam in these experiments (~ 15  $\mu$ m in diameter). Being in an open system with a constant flow of oxygen and ozone, we cannot rule out any mass loss occurring during these experiments. Compared with the effect of particle phase state, we do not expect nonanal loss to impact significantly on the reaction rate.]

We assume that the reviewer refers to the Raman spectra for the levitated particles. The Raman spectrum before and after ozonolysis is included in the Supporting Information file.

**6) In section 3.3.3 it is stated that the weak shoulder at ~2854 cm-1 becomes more defined during oxidation and that this indicates the oleic acid left is not involved in the acid-soap structure. Please include what this shoulder corresponds to, how does the increase in definition show this?**

The shoulder at ~2854 cm-1 corresponds to the  $-CH_2$  symmetric stretching band in the free oleic acid, as opposed to the asymmetric stretching band of the ordered alkyl chains (~2887 cm-1) (Tandon et al., 2001). There is a molar excess of oleic acid in the mixture, so this is to be expected. As ozonolysis proceeds this shoulder becomes a more defined peak. This area of the Raman spectrum now resembles that of liquid oleic acid. This spectrum is presented in the Supporting Information file along with a comparison with the other components of the mixture. We have amended the text accordingly to define these two bands more clearly.

[Three key changes in the Raman spectrum are observed during ozonolysis. First, there is a clear shift of the strong acid—soap peak from ~ 2887 cm-1 (-CH2 asymmetric stretching band) to ~ 2897 cm-1 accompanied by some broadening.]

And:

[Secondly, the weak shoulder at ~ 2854 cm-1 (-CH2 symmetric stretch) becomes a more defined **peak** during oxidation. This region of the Raman spectrum resembles that of **oleic acid (Fig. S1(c))**. This is further evidence, in combination with SAXS observations (Fig. 2(j)), that the oleic acid left in the system is not involved in an acid–soap structure after ozonolysis.]

7) Is there data for the evolution of the Raman spectra during ozonolysis? I could not find this in the supplemental and it would be nice to include, especially given that changes in peaks are discussed.

We thank the reviewer for this comment and we have now included the raw Raman spectra data for ozonolysis of the acid-soap complex in the .zip file accompanying the Supporting Information document. These data backup Fig. 2(j) in the main text and Fig. S6 in the Supporting Information, which show the evolution of the Raman spectrum during ozonolysis.

**8) Page 17 discusses inverse micellar vs. micellar. Please provide a little more information on why inverse micelles are expected for this system, even at the highest RH values.**

We thank the reviewer for this comment and understand the need for extra clarity on this point. We have added the following to the paragraph discussing this in section 4:

[...It is therefore likely that the micellar phase observed in this system has an inverse rather than the normal topology suggested in the preceding atmospheric literature (see Fig. 3 for a cartoon representation). The larger hydrophobic tail region of the oleic acid molecule compared with its hydrophilic head drives interface curvature towards water, and therefore the formation of inverse phases, even in excess water. Inverted micellar and other inverted topology phases have been observed for oleic acid – sodium oleate mixtures in excess water (Seddon et al., 1990). Normal topology micelles (polar head groups at the micelle surface) form in systems with larger / charged headgroups, and are only observed within the sodium oleate – oleic acid system at high sodium oleate content (> 80 wt %) (Seddon et al., 1990). It is therefore likely that the micellar phase observed in this system has an inverse rather than normal topology suggested in the preceding atmospheric literature (see Fig. 3 for a cartoon representation).]

**9) The data provided in the figures in the manuscript look like data for a single experiment. Were replicate experiments run for these samples? If so, which ones and how many replicates?**

The ozonolysis and humidity experiments were carried out in different particles. The vertical and horizontal radii are quoted in the figure captions for Fig. 1 and 2. We recognise that in the methods section 2.3 the final paragraph makes reference to the size of the particles but does not state the number studied. One particle was studied for the humidity experiment and three for the ozonolysis experiments (including the oleic acid particle) – see Fig. 2(i) in the main text.

The methods section 2.3 has been updated:

[The levitated particles were analysed on the I22 beamline at the Diamond Light Source (UK). Solid samples, crystallised from ethanolic solutions, were placed into a node of the acoustic levitator. The particles had vertical radii of ~ 90–150  $\mu$ m and horizontal radii of ~ 500  $\mu$ m (determined using the attenuation of the X-ray beam). One particle of proxy mixture was levitated subjected to the humidity change experiment. Two particles of the proxy mixture and one pure oleic acid particle were levitated and subjected to ozonolysis experiments.]

The figure caption for Fig. 2 has also been updated for extra clarity:

[Figure 2. Vertical scans through the particle showing the effect of ozonolysis on selfassembly. Each row of plots ((a) and (b), (c) and (d), (e) and (f), (g) and (h)) shows simultaneous 1–D SAXS and WAXS scattering patterns vs. distance from the particle centre (measured in µm from what was deemed the particle centre from attenuation data) at increasing time exposed to ozone (labelled at the top-right of every WAXS plot). The particle moved and possibly changed shape during the experiment, vertical movement is apparent from the SAXS and WAXS patterns. (i) Comparison of a levitated pure oleic acid droplet vs. a levitated acid–soap complex particle undergoing ozonolysis, measured by Raman microscopy - A longer ozonolysis experiment on a different levitated acid–soap complex particle is also presented, **totalling two ozonolysis experiments on this proxy**. (j) Evolution of the Raman spectra between 2750 and 3050 cm-1 of a levitated acid–soap complex complex during ozonolysis. (Particle size: ~ 85 µm (vertical radius) x ~ 500 µm (horizontal radius);  $[O3] = 51.9 \pm 0.5$  ppm).]

An extra sentence has been added to the conclusion which clarifies that the acid-soap complex was also observed on microscope slide deposits:

[The oleic acid/sodium oleate acid–soap complex has been identified in an unsaturated fatty acid aerosol proxy. Raman and IR spectroscopy, along with SAXS/WAXS, were used to confirm the formation of the acid–soap complex in acoustically levitated particles. The acid–soap complex was also identified by Raman microscopy on microscope slide deposits.]

**Additional change:**

We noticed a typo on line 300 referring to a "Fig. 6". We have amended this to read "Fig. 1(d) and (f)".

**References**

Lee, J. W. L., Carrascón, V., Gallimore, P. J., Fuller, S. J., Björkegren, A., Spring, D. R., Pope, F. D. and Kalberer, M.: The effect of humidity on the ozonolysis of unsaturated compounds in aerosol particles, Phys. Chem. Chem. Phys., 14(22), 8023–8031, doi:10.1039/c2cp24094g, 2012.

Seddon, J. M., Bartle, E. A. and Mingins, J.: Inverse cubic liquid-crystalline phases of phospholipids and related lyotropic systems, J. Phys. Condens. Matter, 2, SA285–SA290, doi:10.1088/0953-8984/2/S/043, 1990.

Tandon, P., Raudenkolb, S., Neubert, R. H. H., Rettig, W. and Wartewig, S.: X-ray diffraction and spectroscopic studies of oleic acid-sodium oleate, Chem. Phys. Lipids, 109(1), 37–45, doi:10.1016/S0009-3084(00)00207-3, 2001.

Vesna, O., Sax, M., Kalberer, M., Gaschen, A. and Ammann, M.: Product study of oleic acid ozonolysis as function of humidity, Atmos. Environ., 43(24), 3662–3669, doi:10.1016/j.atmosenv.2009.04.047, 2009.